# The structure of EXTL3 helps to explain the different roles of bi-domain exostosins in heparan sulfate synthesis

L. F. L. Wilson [1,6], T. Dendooven [2,7], S. W. Hardwick [2], A. Echevarría-Poza[1], T. Tryfona [1], K. B. R. M. Krogh[3], D. Y. Chirgadze [2], B. F. Luisi [2], D. T. Logan [4], K. Mani [5✉] & P. Dupree [1✉]

Heparan sulfate is a highly modified O-linked glycan that performs diverse physiological roles in animal tissues. Though quickly modified, it is initially synthesised as a polysaccharide of alternating β-D-glucuronosyl and N-acetyl-α-D-glucosaminyl residues by exostosins. These enzymes generally possess two glycosyltransferase domains (GT47 and GT64)—each thought to add one type of monosaccharide unit to the backbone. Although previous structures of murine exostosin-like 2 (EXTL2) provide insight into the GT64 domain, the rest of the bi-domain architecture is yet to be characterised; hence, how the two domains co-operate is unknown. Here, we report the structure of human exostosin-like 3 (EXTL3) in apo and UDP-bound forms. We explain the ineffectiveness of EXTL3's GT47 domain to transfer β-D-glucuronosyl units, and we observe that, in general, the bi-domain architecture would preclude a processive mechanism of backbone extension. We therefore propose that heparan sulfate backbone polymerisation occurs by a simple dissociative mechanism.

[1] Department of Biochemistry, University of Cambridge, Cambridge CB2 1QW, UK. [2] Department of Biochemistry, University of Cambridge, Cambridge CB2 1GA, UK. [3] Department of Protein Biochemistry and Stability, Novozymes A/S, Krogshøjvej 36, 2880 Bagsværd, Denmark. [4] Biochemistry and Structural Biology, Centre for Molecular Protein Science, Department of Chemistry, Lund University, SE-221 00 Lund, Sweden. [5] Department of Experimental Medical Science, Division of Neuroscience, Glycobiology Group, Lund University, SE-221 00 Lund, Sweden. [6] Present address: Department of Molecular Physiology and Biological Physics, University of Virginia, Charlottesville, VA 22903, USA. [7] Present address: MRC Laboratory of Molecular Biology, Francis Crick Avenue, Cambridge CB2 0QH, UK. ✉email: katrin.mani@med.lu.se; pd101@cam.ac.uk

Heparan sulfate proteoglycans (HSPGs) are proteins decorated with heparan sulfate (HS) carbohydrate moieties[1]. HSPGs play a variety of important roles in animal tissues, including acting as cell surface receptors, modulating enzyme activities, and fulfilling structural roles in the extracellular matrix[1–3]. These functions are primarily achieved through the ability of HS to bind avidly to a very wide range of interactors—including growth factors, cytokines, and the spike protein of SARS-CoV-2[1,3,4]. The fundamental importance of HS is reflected in its invariant presence in animal extracellular spaces[3], as well as its conservation in metazoans from Cnidaria to vertebrates[5,6].

Along with chondroitin sulfate/dermatan sulfate (CS/DS), hyaluronic acid, and other similar polymers, HS is a glycosaminoglycan (GAG): a polysaccharide of alternating hexosamine and hexose/hexuronic acid residues[7]. Most GAGs are covalently linked to a core protein; such conjugates are known as proteoglycans[2]. In particular, HS and CS/DS chains are $O$-linked to serine residues in the core protein via a tetrasaccharide linker with structure -4-GlcA-β1,3-Gal-β1,3-Gal-β1,4-Xyl-β1-[1]. HS is initially synthesised as a polysaccharide of alternating α1,4-linked $N$-acetylglucosamine (α-GlcNAc) and β1,4-linked glucuronic acid (β-GlcA) sugars. Concomitantly with chain polymerisation, the HS chain undergoes modifications at various positions including $N$-deacetylation/$N$-sulfation, epimerization, 2-$O$-sulfation, 6-$O$-sulfation, and 3-$O$-sulfation, resulting in the enormous structural diversity of these macromolecules[7,8].

The HS backbone is synthesised in the Golgi apparatus by enzymes known as exostosins[8–10]. The human genome encodes five exostosins: exostosin 1 (EXT1), exostosin 2 (EXT2) and exostosin-like 1–3 (EXTL1–3). Most exostosins contain two glycosyltransferase domains; it has been proposed that each domain is responsible for one of the two types of linkage in the nascent HS polysaccharide[9,11,12]. The N-terminal domain, which belongs to CAZy[13–15] family GT47, is thought to be capable of adding β-GlcA[16,17], while the C-terminal domain belongs to the GT64 family and is capable of adding α-GlcNAc[18]. The HS chain can be initiated from the tetrasaccharide linker by EXTL3 or (in some cases) by EXTL2, which add the first α-GlcNAc residue (GlcNAcT-I activity)[8,9]. Importantly, this step determines that the tetrasaccharide will become decorated with HS, and not CS/DS[19]. From here, the addition of β-GlcA and further α-GlcNAc residues (GlcAT-II and GlcNAcT-II activity, respectively) is achieved mainly by EXT1 and EXT2, which are thought to operate as a hetero-oligomer[9,20]. EXTL3 and EXTL1 are also able to transfer α-GlcNAc residues during elongation (GlcNAcT-II); however, both have been reported to lack GlcAT-II activity[21,22]. Recent evidence now indicates that EXTL2, which lacks a GT47 domain[13,14], is involved in blocking the initiation of HS and CS/DS chains by adding α-GlcNAc to the linkage tetrasaccharide before an inhibitory phosphate can be removed from the reducing-end xylose[23].

It is frequently reported that pathway-related Golgi enzymes are physically co-localised[24–26]. Furthermore, a number of tandem (or 'bi-domain') glycosyltransferase fusions have been identified in the Golgi—including exostosins (GT47–GT64), chondroitin synthases (GT31–GT7), and LARGE enzymes (GT8–GT49), all of which are thought to synthesise alternating polysaccharides[9,13,14,27,28]. However, because no such enzyme complex or fusion has yet been structurally characterised, the function of this physical clustering is unclear. Therefore, the bi-domain members of the exostosin family represent an interesting target for structural investigation. So far, only the sole GT64 domain of the smallest exostosin, EXTL2, has been crystallised[29]. The crystal structure revealed that EXTL2 exists as a symmetric homodimer, and that the GT64 domain adopts a GT-A fold, binding UDP-GlcNAc or UDP-GalNAc with the aid of a

manganese cofactor. While the GT47 domain of the larger exostosins has been tentatively predicted to adopt a GT-B fold[30], the overall bi-domain structure remains to be determined, and the GT47 fold is yet to be described. This missing information could provide insight into the mechanism of HS chain extension and the diverse activities of GT47 family members in plants[17,31].

Although humans produce five exostosins, only EXT1, EXT2, and EXTL3 are conserved across the animal kingdom[5,9], suggesting that these enzymes encompass the core elements required for HS backbone synthesis. Unlike EXT1 and EXT2, whose activities are reported to vary according to hetero-oligomeric state[20,32], EXTL3 is likely homomeric and not known to require any additional factor for complete activity, simplifying structure-function analyses. We recently described a low-resolution SAXS structure of EXTL3, the largest exostosin[30]. Here, we present high-resolution cryo-EM structures of the 170 kDa globular portion of an EXTL3 homodimer in the apo-form (2.4 Å resolution) and bound with UDP and $Mn^{2+}$ (2.9 Å). We locate the active sites, describe the GT47 fold, explain the loss of GlcAT-II activity in EXTL3 compared with other exostosins, and speculate as to how EXTL3 achieves specificity for HS addition sites. Together, these results help to explain some of the molecular mechanisms for genetic diseases caused by mutations in exostosin genes (such as hereditary multiple exostoses and spondylo-epimetaphyseal dysplasia) and provide insight into the organisation of glycosylation reactions within the Golgi.

## Results

**A sensitive assay can detect GlcNAcT-II and weak GlcAT-II activity in EXTL3ΔN preparations.** We previously created a soluble form of human EXTL3 lacking the first 51 amino acids from the N terminus (EXTL3ΔN)[30]. This protein contains a predicted coiled coil domain[33], a GT64 glycosyltransferase domain, and a GT47 glycosyltransferase domain, but lacks a transmembrane helix (Fig. 1a). To confirm that this form of EXTL3 is catalytically active, we expressed EXTL3ΔN in human embryonic kidney (EBNA 293) cells and purified the protein from the cell culture medium by Ni-NTA and size exclusion chromatography. Label-free quantification (LFQ) mass spectrometry experiments indicated that the abundances of endogenous EXT1 and EXT2 were 10,000–100,000-fold less than that of EXTL3ΔN; furthermore, EXTL1 and EXTL2 were not detected (Supplementary Tables 1 and 2). Hence, we proceeded under the assumption that the activity in these preparations originated from EXTL3ΔN.

Next, we measured the $N$-acetylglucosaminyltransferase activity of EXTL3ΔN. Since we were unable to obtain an appropriate acceptor for measuring GlcNAcT-I initiation activity (the primary activity of EXTL3), we instead observed the GlcNAcT-II activity of EXTL3ΔN through its ability to extend K5 heparosan oligosaccharides, which have an identical structure to the nascent HS backbone and are a well-established substrate in exostosin activity assays[11,34]. Accordingly, we incubated purified EXTL3ΔN with UDP-GlcNAc and a DP8 (degree of polymerisation = 8) K5 acceptor ([GlcA-GlcNAc]$_4$) terminating in β-GlcA. The products were then characterised by polysaccharide analysis by carbohydrate electrophoresis (PACE). Consistent with the reported activity of EXTL3, the [GlcA-GlcNAc]$_4$ acceptor was completely converted to the DP9 oligosaccharide GlcNAc-[GlcA-GlcNAc]$_4$ following overnight incubation (Fig. 1b). No GlcNAcT-II activity was observed when EXTL3ΔN and/or UDP-GlcNAc were omitted, or when $MnCl_2$ and $MgCl_2$ were replaced with EDTA. Taken together, these results indicate that our preparations of EXTL3ΔN exhibit a metal-dependent GlcNAcT-II activity.

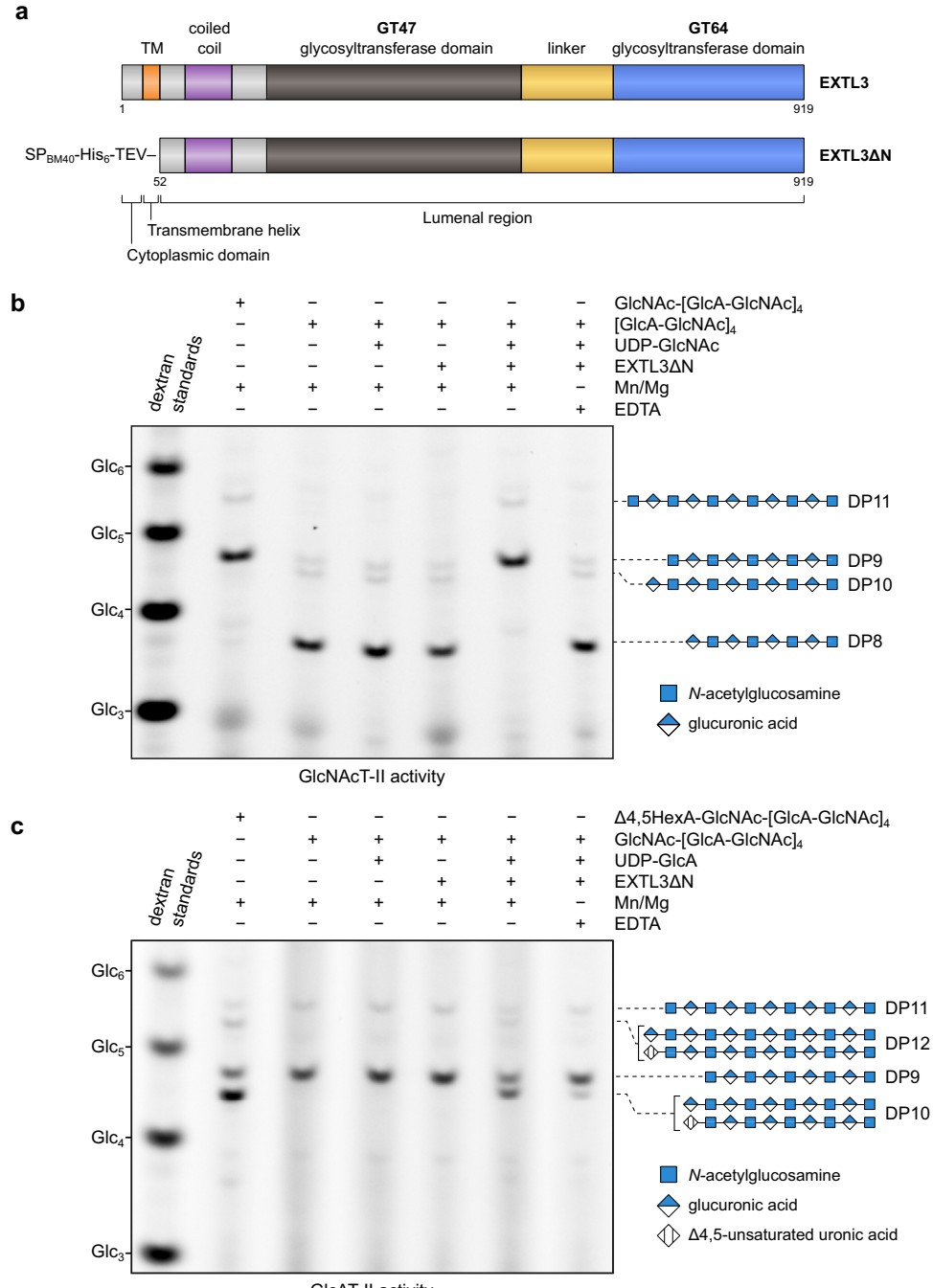

**Fig. 1 Carbohydrate electrophoresis can be used to observe heparosan-extending activity in preparations of EXTL3ΔN, a secretory form of EXTL3.**
**a** Domain annotation of full-length EXTL3 and EXTL3ΔN, respectively. **b, c** Polysaccharide analysis by carbohydrate electrophoresis (PACE) gels showing reaction products of EXTL3ΔN GlcNAcT-II and GlcAT-II assays, respectively. Secretory EXTL3ΔN was expressed in human embryonic kidney cells and purified from the culture medium. Acceptors (DP8 [GlcA-GlcNAc]₄ or DP9 GlcNAc-[GlcA-GlcNAc]₄) were prepared enzymatically from the *E. coli* K5-derived oligosaccharide Δ4,5HexA-GlcNAc-[GlcA-GlcNAc]₄ (DP10). The DP8 and DP9 acceptor preparations also contained a small amount of DP10 [GlcA-GlcNAc]₅ or DP11 GlcNAc-[GlcA-GlcNAc]₅ oligosaccharide, respectively. For each assay, acceptor was incubated with a combination of EXTL3ΔN, UDP-GlcNAc, UDP-GlcA, MnCl₂ and MgCl₂, and/or EDTA (as indicated) for 16 h. The products were subsequently derivatised with a fluorophore and separated by PACE. Results representative of two biologically independent experiments.

Although our mass spectrometry experiments indicated that the levels of EXT1/2 were very low, we also assayed our EXTL3ΔN preparations for potential GlcAT-II activity. Accordingly, we incubated the preparation of EXTL3ΔN with UDP-GlcA and a DP9 K5 acceptor (GlcNAc-[GlcA-GlcNAc]₄) terminating in α-GlcNAc. Surprisingly, we detected appreciable, albeit incomplete, conversion of the DP9 acceptor to a putative DP10

[GlcA-GlcNAc]₅ oligosaccharide following overnight incubation (Fig. 1c). Conversely, no conversion of DP9 acceptor to DP10 was observed when EXTL3ΔN and/or UDP-GlcNAc were omitted. However, unlike the GlcNAcT-II activity, the apparent GlcAT-II activity was only partially sensitive to EDTA treatment. To verify the structure of the product, we analysed the reactants and products from a completed reaction by matrix-assisted laser

desorption ionisation–time of flight (MALDI-TOF) mass spectrometry. The mass spectrum indicated a mass increase of 176 Da between acceptor and product, consistent with the addition of a β-GlcA residue (Supplementary Fig. 1a, b). Furthermore, PACE analysis revealed that the product was sensitive to β-glucuronidase digestion (Supplementary Fig. 1d). We also incubated the enzyme and DP9 acceptor in the presence of UDP-GlcNAc instead of UDP-GlcA. In this case, no activity was seen by PACE analysis (Supplementary Fig. 1d). By contrast, when enzyme and DP9 acceptor were incubated simultaneously with UDP-GlcA and UDP-GlcNAc, a ladder of larger products was observed (Supplementary Fig. 1d). The masses of these products were consistent with a series of oligosaccharides with general formula $GlcNAc_{n+1}GlcA_n$, as well as a much smaller population with general formula $GlcA_nGlcNAc_n$ (Supplementary Fig. 1c). These results indicate that, in addition to GlcNAcT-II activity, our preparations of EXTL3ΔN also exhibited a weaker level of GlcAT-II activity.

**The GlcAT-II activity of EXTL3ΔN preparations is reduced when EXT1 is targeted by CRISPR-Cas9.** The EXT1/2 complex has been shown to exhibit very strong GlcAT-II activity, with comparatively little GlcNAcT-II activity[20,32]. As EXTL3 has been previously shown to lack GlcAT-II activity[21,22], we considered the possibility that the GlcAT-II activity in our preparations might in fact originate from trace levels of EXT1/2. To investigate this, we mutated endogenous *EXT1* in EXTL3ΔN-expressing EBNA 293 cells using CRISPR-Cas9. We grew three separate cell cultures: regular EXTL3ΔN-expressing cells (EXTL3ΔN), EXTL3ΔN-expressing cells transfected with a non-specific CRISPR-Cas9 construct (not targeting any known gene; EXTL3ΔN CRISPR^control), and EXTL3ΔN-expressing cells transfected with a CRISPR-Cas9 construct targeting EXT1 (EXTL3ΔN CRISPR^EXT1). Immunofluorescence microscopy and slot blot assays indicated a substantial decrease (by $81 \pm 10\%$ and $67 \pm 25\%$, respectively) in immunoreactive EXT1 in the EXTL3ΔN CRISPR^EXT1 culture compared with the EXTL3ΔN CRISPR^control, though the signal was not totally abolished (Supplementary Figs. 2 and 3).

Subsequently, we collected and purified EXTL3ΔN from the three different cell cultures. After equalising the total protein concentration between the three EXTL3ΔN preparations, we subjected them to LFQ mass spectrometry analysis, as described above. As expected, all three samples exhibited very similar levels of EXTL3ΔN (Fig. 2a and Supplementary Table 3). Consistent with our previous results, EXTL3ΔN preparations from EXTL3ΔN and EXTL3ΔN CRISPR^control cells exhibited levels of EXT1 and EXT2 that were 10,000–100,000-fold lower than those of EXTL3ΔN. By contrast, the abundance of EXT1 in the EXTL3ΔN CRISPR^EXT1 sample was below the threshold for detection. Interestingly, the level of EXT2 was also partially reduced in this sample. Hence, the abundance of contaminating EXT1/2 heteromer was almost certainly diminished in the EXTL3ΔN CRISPR^EXT1 sample.

We then proceeded to quantify the GlcNAcT-II and GlcAT-II activities of the three different preparations. To do so, we quantified the percentage substrate conversion after 15 and 60 min using PACE. Firstly, the results indicated that the GlcNAcT-II activity was generally at least fourfold greater than the GlcAT-II activity (Fig. 2b–e). Furthermore, the GlcNAcT-II activity was essentially invariant between the three preparations (Fig. 2b, d; no significant difference from one-way ANOVA; $p > 0.05$). In contrast, the GlcAT-II activity was inconsistent between all three preparations (Fig. 2c, e; $p < 0.001$), and appeared to correlate with the abundance of EXT1 (as determined by LFQ). Most importantly, the GlcAT-II activity was significantly reduced

in the EXTL3ΔN CRISPR^EXT1 preparation compared to both the EXTL3ΔN preparation (Tukey's HSD; in relative terms, a reduction of $37 \pm 31\%$ after 15 min, $p < 0.01$; and $47 \pm 28\%$, $p < 0.01$, after 60 min) and the EXTL3ΔN CRISPR^control preparation ($55 \pm 23\%$, $p < 0.001$ after 15 min; $63 \pm 20\%$, $p < 0.001$ after 60 min). These results strongly suggest that the observed GlcAT-II activity—but not GlcNAcT-II activity—was dependent on the presence of EXT1. Although some GlcAT-II activity was still present in the EXTL3ΔN CRISPR^EXT1 preparation, this might be attributable to EXT2, or trace amounts of residual EXT1. Hence, given that EXTL3 has previously been found to lack GlcAT-II activity[21,22], and that the EXT1/2 complex is known to exhibit excessively high GlcAT-II activity[20,32], it seems likely that the majority of the observed GlcAT-II activity did not originate from EXTL3ΔN.

**Quaternary architecture of EXTL3.** Our previous SAXS structure of EXTL3 suggests that it forms a homodimer[30], which would include four glycosyltransferase domains. To gain further insight into the arrangement of these domains, we solved the structure of EXTL3ΔN by single-particle cryo-EM, producing a high-quality Coulomb potential map at 2.4 Å resolution (Fig. 3 and Supplementary Figs. 4–6). We then built and refined an ab initio model of the 170 kDa globular domain of an EXTL3 homodimer (Fig. 3). In each chain, we were able to model two partial *N*-glycans at Asn592 and Asn790, respectively (Supplementary Fig. 7). The coiled coil domain, however, was not visible in the map, likely due to its flexibility with respect to the catalytic domain.

Detailed analysis of the globular structure revealed that it is indeed composed of four glycosyltransferase-type domains, with one GT47 and GT64 domain provided by each chain. Within each chain, the GT47 and GT64 domains are connected by a linker region of 124 amino acids. Curiously, the N-terminal half of the linker region constitutes an extended loop that traverses 40 Å over the surface of the GT64 domain, forming a 'cradle' (Supplementary Fig. 8). The GT64 domain and linker region appear to contribute the most to the homodimeric interaction (see Supplementary Table 4 for PISA[35]-calculated energy contributions), which appears to be stabilised by a pair of intermolecular disulfide bridges between Cys793 and Cys915 in the GT64 domain (Fig. 3).

**GT64 domain structure.** The reported crystal structure of mouse EXTL2 constitutes a GT64-domain homodimer, with no GT47 domain or linker region[29]. We aligned the GT64 portion of EXTL3 with the apo-EXTL2 structure (PDB: 1OMX), finding a good agreement, with an RMSD of 0.94 Å between the 161 aligned $C_\alpha$ atoms of the monomeric EXTL2 and EXTL3 GT64 domains (Fig. 4a). Furthermore, the homodimerisation of the EXTL3 GT64 domain, which involves the formation of an intermolecular β-sheet, highly resembles that of EXTL2, suggesting that this mode of interaction may also be conserved in other exostosins (Fig. 4b).

In contrast, the EXTL3 GT64 domain differs substantially from EXTL2 in the structure of the C-terminal loop (Fig. 4a). A further structural alignment with acceptor-analogue-bound EXTL2 (PDB: 1ON8; Fig. 4c) revealed that the loop sits proximally to the GT64 acceptor binding site. Whereas in EXTL2 this loop is relatively short and terminates in the solvent, in EXTL3 the loop is considerably longer and sees the C-terminus embedded in the protein. Furthermore, the EXTL3 loop is rich in cationic sidechains and contains a solvent-exposed phenylalanine side chain (Phe918; Fig. 4c). These observations are interesting in light of the previous proposition that EXTL3 might recognise the

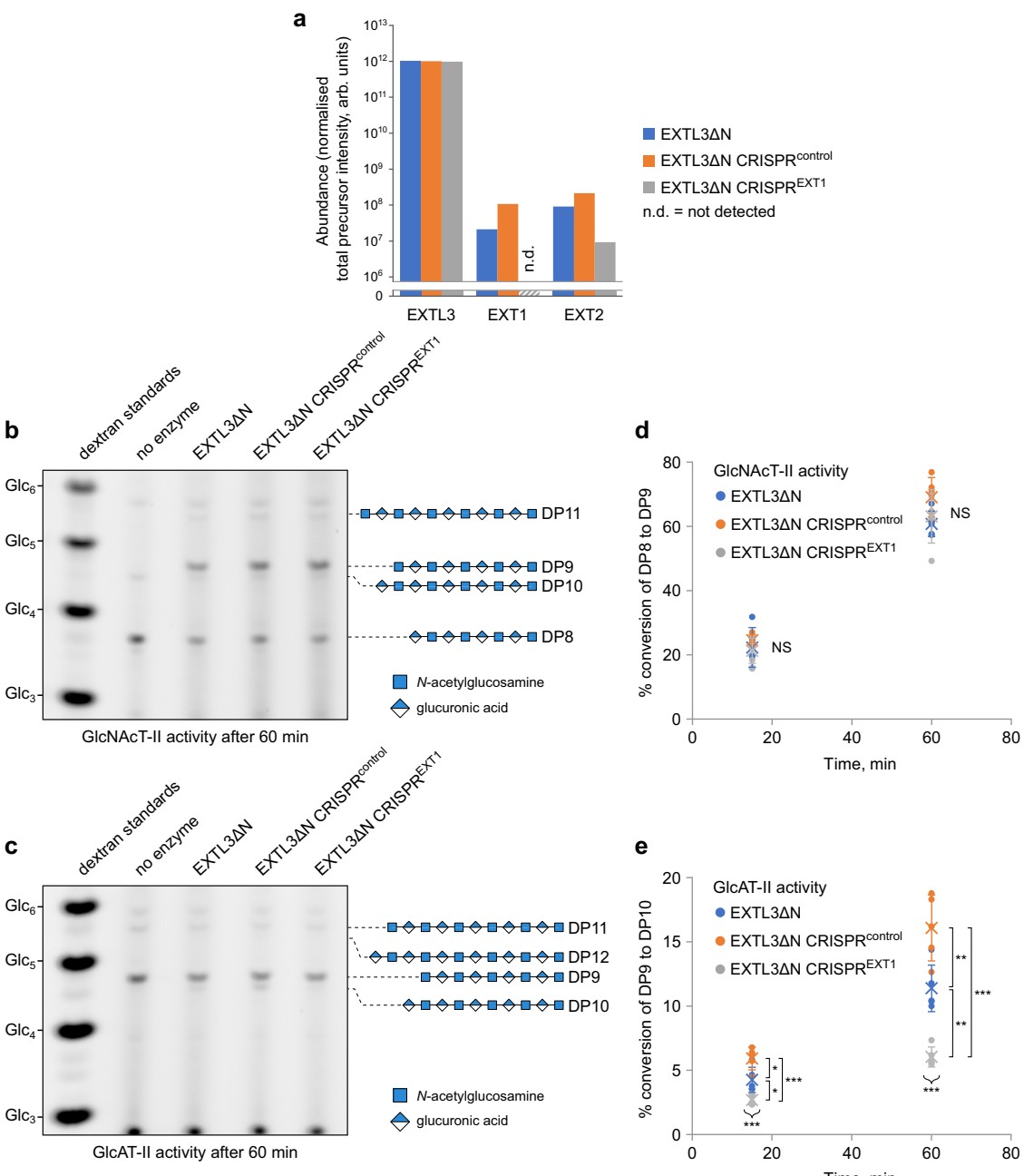

**Fig. 2 The level of GlcAT-II activity in EXTL3ΔN preparations exhibits substantial dependence on the level of EXT1 contamination.** EXTL3ΔN was prepared from the culture medium of regular EXTL3ΔN-expressing cells (EXTL3ΔN), EXTL3ΔN-expressing cells transfected with a non-specific CRISPR-Cas9 control construct (EXTL3ΔN CRISPR^control), or EXTL3ΔN-expressing cells transfected with a CRISPR-Cas9 construct targeting EXT1 (EXTL3ΔN CRISPR^EXT1). **a** The relative abundance of EXTL3(ΔN), EXT1, and EXT2 in the three different preparations was determined by mass spectrometry/label-free quantification. The lower limit of detection corresponds to approximately $10^6$ arbitrary units. For numerical values, as well as the top ten most abundant proteins, see Supplementary Table 3. Each dataset was obtained from a single biological replicate. **b–e** Comparison of GlcNAcT-II and GlcAT-II activities of EXTL3ΔN, EXTL3ΔN CRISPR^control, and EXTL3ΔN CRISPR^EXT1 preparation, visualised by PACE. For both activities, five independent assays were carried out using the same biological enzyme preparation (i.e. $n = 5$ technical replicates for each treatment) to account for potential technical variability. **b**, **d** GlcNAcT-II activity: DP8 acceptor was treated with enzyme in the presence of UDP-GlcNAc, MnCl$_2$, and MgCl$_2$. **c**, **e** GlcAT-II activity: DP9 acceptor was treated with enzyme in the presence of UDP-GlcA, MnCl$_2$, and MgCl$_2$. Reactions were then split in two; each half was terminated after 15 or 60 min, respectively. **b**, **c** Representative PACE gels for GlcNAcT-II and GlcAT-II 60 min time points, respectively. **d**, **e** Percentage substrate conversion at the two different time points was determined by densitometry. Dots represent individual measurements, crosses represent group means, and error bars represent standard error of the mean. Statistics: one-way ANOVA (two-tailed) with Tukey's honest significant difference (HSD); *NS* no significant difference * $p < 0.05$, ** $p < 0.01$, *** $p < 0.001$. GlcNAcT-II ANOVA: $F_{2,12} = 0.75$, $p = 0.49$ (15 min); $F_{2,12} = 2.23$, $p = 0.15$ (60 min). GlcAT-II ANOVA: $F_{2,12} = 21.09$, $p = 1.18 \times 10^{-4}$ (15 min); $F_{2,12} = 36.2$, $p = 8.29 \times 10^{-6}$ (60 min). GlcAT-II Tukey's HSD between EXTL3ΔN (1), EXTL3ΔN CRISPR^control (2), and EXTL3ΔN CRISPR^EXT1 (3): $p = 0.0149$ (1-2), $p = 0.0213$ (1-3), $p = 8.15 \times 10^{-4}$ (2-3) (15 min); $p = 0.0479$ (1-2), $p = 0.00187$ (1-3), $p = 5.6 \times 10^{-6}$ (2-3) (60 min).

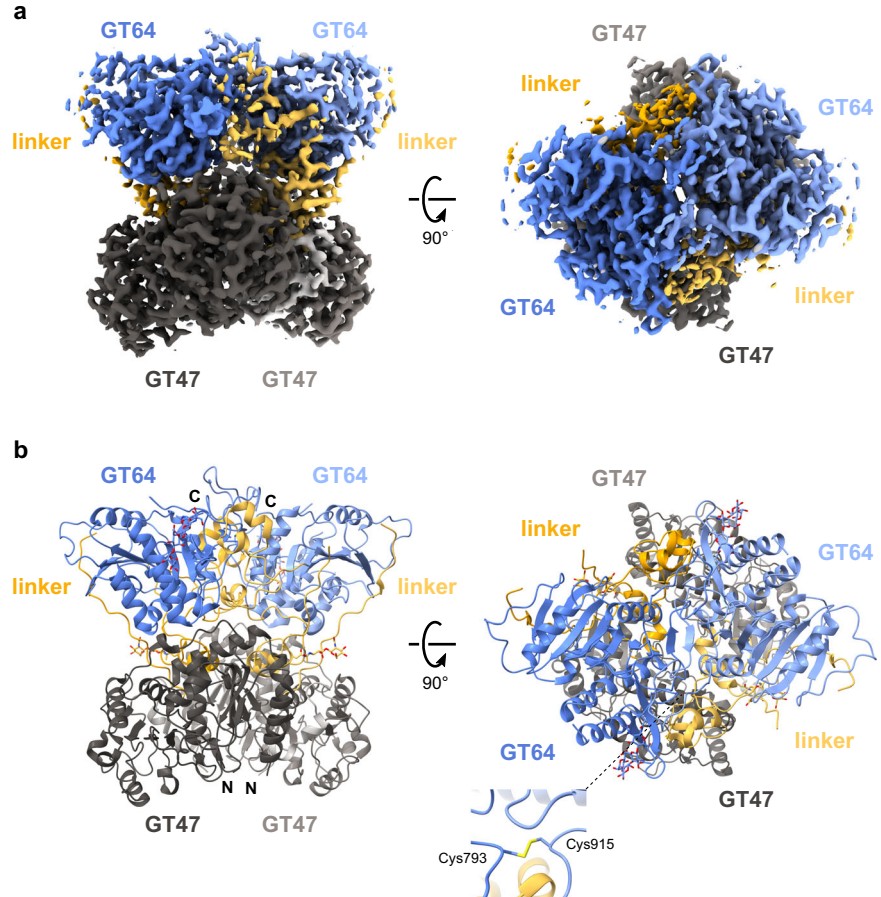

**Fig. 3 Cryo-EM structure of the homodimeric globular domain of EXTL3.** EXTL3ΔN was purified and subjected to cryo-EM single particle analysis, achieving a resolution of 2.4 Å. **a** Cryo-EM density map. Domains are coloured-coded as in Fig. 1a (GT47: dark grey; inter-domain linker: yellow; GT64 blue). **b** Fitted model, coloured-coded as in panel a. The N- and C-termini are labelled for each chain.

anionic and hydrophobic residues that commonly surround HS sites in core proteins[36].

**GT47 domain structure**. Inspection of the structure revealed that the EXTL3 GT47 domain adopts a GT-B fold, with two distinct Rossman-fold subdomains (Fig. 5a). However, the EXTL3 GT47 domain appears to constitute a minimal GT-B fold, with only five β-strands in each subdomain, and only eight substantial α-helices in the entire fold (Fig. 5b, c). Using the DALI server[37] to detect structural homologues, we identified glycosyltransferases from GT-B families GT90, GT63, GT113, GT4, and GT72 as the five most similar matches (Supplementary Table 5). Notably, the structure of GtfC from *Streptococcus agalactiae* (PDB: 4W6Q; GT113) was also identified as a close match when we searched using only the N-terminal or C-terminal subdomains, suggesting that the bacterial GT113 family might be closely related to the eukaryotic GT47 family.

Unlike GT-A-fold glycosyltransferases, which commonly possess a DxD metal-binding motif, GT-B-fold enzymes are not thought to contain any widespread amino acid motifs[38,39]. Nevertheless, at least some GT-B enzymes exhibit a conserved aspartate/glutamate residue in the fourth α-helix of the C-terminal subdomain (Cα4);[40] the side chain of this residue typically forms a hydrogen bond with the ribose of the nucleotide sugar[38,41]. To investigate whether such a pattern might be present in EXTL3, we examined the GT47 structural alignments produced by DALI. Indeed, in the EXTL3 GT47 domain, Glu453 aligns well with similarly placed acidic residues in other

GT-B structures (Fig. 5d). To examine the conservation of this residue amongst the wider GT47 family, we aligned the EXTL3 GT47 domain sequence with all other characterised GT47 glycosyltransferases from *Homo sapiens* and *Arabidopsis thaliana*. Indeed, Asp/Glu residues were strongly conserved at this position, supporting the idea that these residues are involved in binding the nucleotide sugar in those enzymes (Fig. 5e).

In spite of this sequence conservation, the GT47 domain of EXTL3 exhibits several unique features that point to a loss of activity in this particular enzyme. For instance, the side chains of Glu453 and Arg421 are engaged in a salt-bridge, which would direct Glu453 away from making interactions with the substrate (Fig. 5d). Moreover, our structural alignment revealed that, in contrast to other GT-B structures, the EXTL3 Cα4 helix exhibits an extra turn at its N-terminus that occludes a putative phosphate/donor sugar binding pocket (Fig. 5d). In the other substrate-bound GT-B structures that we examined, the phosphates and ribose moiety of the donor sugar were consistently bound in this pocket, as has been previously documented for GT-B-fold and other Rossmann-fold enzymes[38,42]. Consultation of the GT47 domain alignment revealed that both of these unusual features are absent from other characterised GT47 enzymes: no analogue to Arg421 was present in other GT47s (except in the case of EXTL1, which is not known to possess GlcAT-II activity), and the conspicuous pre-helical insertion was not observed in any of the other exostosin or plant GT47 sequences (Fig. 5e).

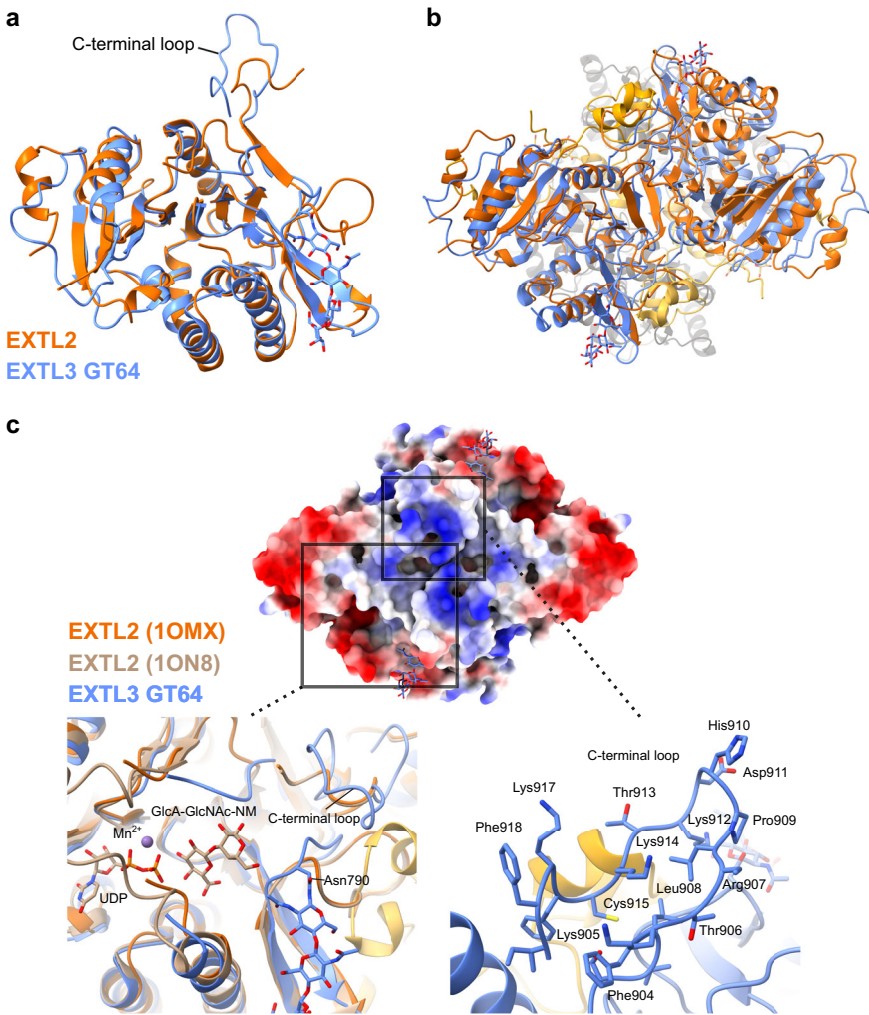

**Fig. 4 The GT64 domain of EXTL3 shows high similarity with that of EXTL2 from mouse, but differs substantially in the C-terminal loop. a** Individual EXTL3 GT64 domain (blue) aligned to a single murine apo-EXTL2 monomer (PDB: 1OMX; orange). **b** Complete homodimeric EXTL2 structure (orange) aligned to homodimeric EXTL3 (coloured as in Fig. 3) via an individual EXTL2 GT64 domain. **c** Top surface of EXTL3, coloured by electrostatic potential (blue: most positive; red: most negative). Bottom left: Alignment of murine apo-EXTL2 (orange), EXTL3 (coloured as in Fig. 3), and murine EXTL2 bound with UDP and an acceptor analogue mimicking the final two residues of the tetrasaccharide linker (GlcA-β1,3-Gal-β1-*O*-naphthalenemethanol; 'GlcA-GlcNAc-NM') (PDB: 1ON8; tan), showing the GT64 acceptor binding cleft. The visible C-terminal loop in EXTL3 is provided by the opposing monomer. Bottom right: Close-up of the C-terminal loop. The side chain of Arg907 was not modelled due to lack of density.

During manuscript preparation, high-accuracy structural predictions were made available for the human exostosins at the AlphaFold Protein Structure Database[43,44] (https://alphafold.ebi.ac.uk/). Comparison of the aligned structures indicated that, whereas both the experimental and predicted EXTL3 structures exhibit a four-turn Cα4 helix and Glu453–Arg521 salt bridge, both EXT1 and EXT2 models exhibit a more canonical three-turn Cα4 helix and no salt bridge, providing space for nucleotide sugar binding (Supplementary Fig. 9). Supporting the hypothesis that UDP-GlcA binds to this site in EXT1, numerous point mutations that inactivate GlcAT-II activity in Chinese hamster EXT1[16] align to positions around the N-terminus of its predicted Cα4 helix (Supplementary Fig. 10). These include a mutation to Glu349 (the equivalent to Glu453 in EXTL3). Furthermore, nearby Arg340 (which lacks an equivalent in EXTL3) is the most commonly mutated EXT1 residue in hereditary multiple exostoses[45–47]. Hence, it seems that the unique features of EXTL3's active site likely disrupt UDP-GlcA binding and must have occurred relatively recently in the evolution of the GT47 family.

**UDP binds in the GT64 domain but not in the GT47 domain.** EXTL3 is thought to possess both GlcNAcT-I and GlcNAcT-II activities[9]. The location(s) of the corresponding UDP-GlcNAc-binding active site(s) within EXTL3 are unknown—although analogy with EXTL2 suggests that the GlcNAcT-I activity, at least, is catalysed by the GT64 domain. Furthermore, the presence of GlcAT-II activity in our EXTL3ΔN preparations presented the possibility that EXTL3 possesses an additional UDP-GlcA-binding active site. In structural studies, the active sites of nucleotide-sugar-utilising glycosyltransferases are routinely located through co-crystallisation with the relevant donor nucleotide[48], which binds stably to the active site. Hence, to locate the positions of all potential active site(s) simultaneously, we used single-particle cryo-EM to solve the structure of EXTL3 in the presence of UDP and MnCl$_2$, achieving a resolution of 2.9 Å (Supplementary Figs. 11–13).

In the GT64 domain of the substrate-bound map, new density corresponding to UDP and a Mn$^{2+}$ ion was clearly visible (Supplementary Fig. 14). The interactions between EXTL3 and UDP are highly similar to those between mouse EXTL2 and UDP-GlcNAc (PDB: 1ON6; Fig. 6a). The uracil base appears to

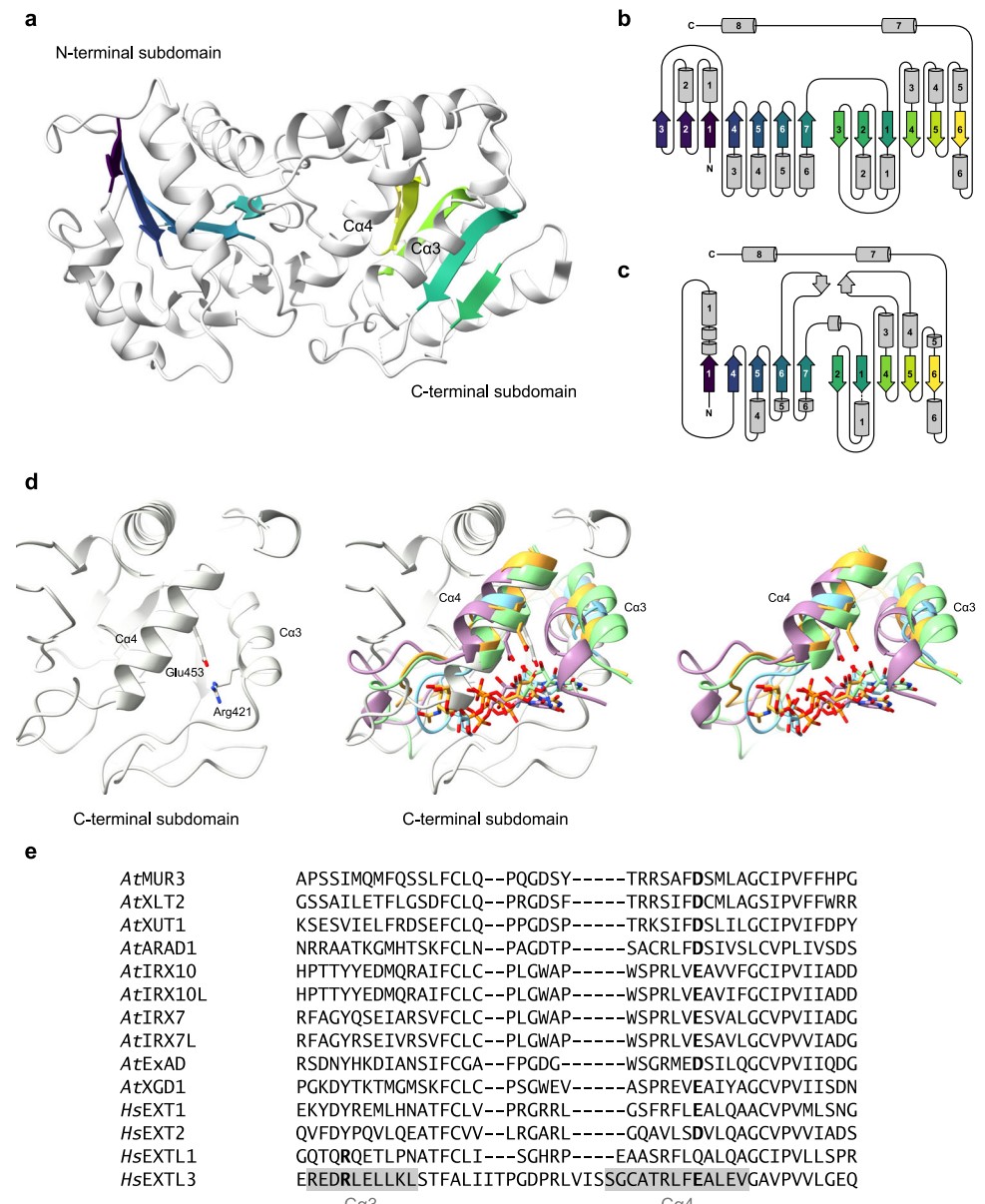

**Fig. 5 The EXTL3 GT47 domain adopts a GT-B fold but appears to lack an appropriate nucleotide sugar binding pocket. a** EXTL3 GT47 domain (from one individual chain). Strands in the central sheets are coloured purple to yellow in order of most N-terminal to most C-terminal. **b** Secondary structure schematic of a generic GT-B-fold glycosyltransferase. **c** Secondary structure schematic of EXTL3 GT47 domain, showing correspondence of the central strands to the GT-B archetype. **d** Structural alignment of the EXTL3 GT47 domain to the closest four GT-B matches from PDB25 identified using DALI (omitting duplicate family members). Left: C-terminal subdomain of EXTL3 GT47 domain (focussing on the Cα3 and Cα4 helices); right: hits from DALI without EXTL3 (only Cα3–Cα4 portion shown for clarity); centre: merged. Structures, by PDB accession and GT family, from highest to lowest match are: 5F84 (GT90; pale blue), 1IXY (GT63; purple), 4W6Q (GT113; pale green), 6EJI (GT4; orange). **e** Protein sequence alignment of EXTL3 GT47 domain with other GT47 domain-containing proteins from *Homo sapiens* and *Arabidopsis thaliana* (only the Cα3–Cα4 portion is shown for brevity). Arg421 and Glu453 residues from EXTL3, as well as their analogues in other GT47 sequences, are highlighted in bold; the positions of the Cα3 and Cα4 helices in the EXTL3 sequence are highlighted in pale grey.

be held in place by hydrogen bonding to the side chains of Asn697 and Asn723 and parallel-displaced π-stacking with Tyr670, while the ribose likely makes hydrogen bonds with the main-chain carbonyl group of Leu668 and the side chain of Asp745. The sidechain of Arg672 interacts with the α-phosphate. The $Mn^{2+}$ ion is co-ordinated by the side chain of Asp746 (one of the DxD aspartates) and both phosphate moieties.

In contrast, despite the relatively high concentration of UDP present during grid freezing (10 mM), thorough examination of the GT47 domain in the UDP-bound map did not reveal any differences to that in the apo-structure map (Supplementary

Fig. 15). This suggests that, as predicted, the EXTL3 GT47 domain does not bind UDP and is therefore unlikely to possess a glycosyltransferase activity.

Nevertheless, we attempted to use the EXTL3 structure to provide insight into the workings of its homologue EXT1, which also exhibits a bi-domain configuration but has retained its GT47 GlcAT-II activity. Since EXT1 possesses both GlcAT-II and GlcNAcT-II activities in the same chain, it is reasonable to propose that these alternating activities might be spatially and/or mechanistically linked for increased efficiency. To evaluate this hypothesis, we estimated the distance between the GT47 and

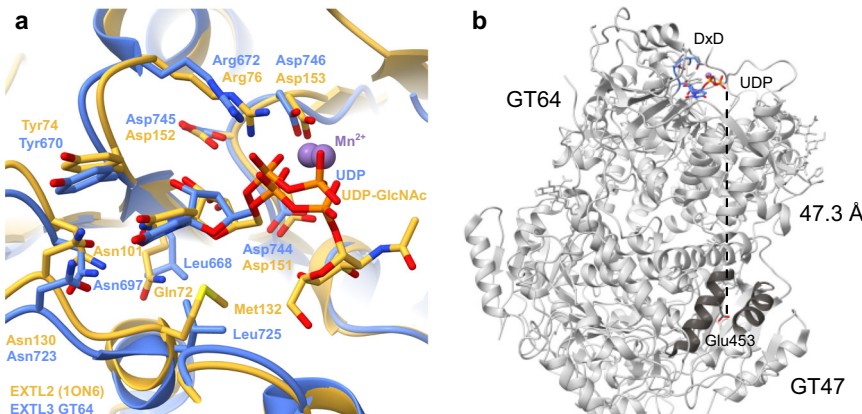

**Fig. 6 UDP binding sites in EXTL3.** Purified EXTL3ΔN was incubated with UDP and subjected to cryo-EM single particle analysis in order to determine its structure. No density for UDP was visible in the GT47 domain. **a** Residues involved in binding UDP and $Mn^{2+}$ in mouse EXTL2 and the GT64 domain of human EXTL3. Human EXTL3 (in complex with UDP and $Mn^{2+}$) is shown in blue (phosphates in orange). Mouse EXTL2 (in complex with UDP-GlcNAc and $Mn^{2+}$; PDB: 1ON6) is shown in tan. **b** The GT64 active site and the apparently inactivated GT47 nucleotide binding site are separated by a substantial distance. The DxD motif and bound UDP of the GT64 domain are shown in blue. The Cα3 and Cα4 helices of the GT47 domain are shown in dark grey. The side chain of Glu453 is shown in stick representation.

GT64 active sites in the EXTL3 structure, which is likely to exhibit the same overall architecture. We observed the distance between the β-phosphate of UDP bound in the EXTL3 GT64 domain and the side chain of Glu453 in the GT47 domain to be over 45 Å (Fig. 6b). Furthermore, a high-confidence model of the EXT1/2 heterodimer generated using the AlphaFold-Multimer[49] Colab server (https://colab.research.google.com/github/deepmind/alphafold/blob/main/notebooks/AlphaFold.ipynb) exhibited a very similar architecture to the EXTL3 homodimer, including the GT64 dimer intermolecular β-sheet seen in EXTL2 and EXTL3 (Supplementary Fig. 16). Importantly, the GT47 and GT64 active sites were separated by almost 50 Å in EXT1—and by over 60 Å in EXT2. Hence, our results indicate that exostosins are highly unlikely to exhibit a processive mechanism of chain extension and must therefore use a simple dissociative mechanism to alternate between the two types of activity.

**The EXTL3 structure rationalises exostosin dysfunction.** Deleterious missense mutations in the EXTL3 gene can cause a range of developmental and neurological disorders in humans. In particular, six amino acid changes have been implicated in disease: P318L, R339W, P461L, R513C, N657S, and Y670D[50–54]. We mapped these mutations onto the EXTL3 structure and deduced that the first five of these mutations likely cause protein destabilisation, as they are distant from the GT64 active site and possess side chains buried in the core of the protein—in several cases connecting potentially flexible loops to other structural elements (Supplementary Table 6 and Supplementary Fig. 17). This is consistent with the fact that the EXTL3[R513C] protein is undetectable in the Golgi bodies of fibroblasts from affected individuals, and is therefore suggested to be mislocalised or degraded[51]. In contrast, our structural data indicated that Tyr670, which relates to the sixth mutation, appears to help bind the uracil moiety of UDP. Its mutation to aspartate therefore likely impairs or inhibits substrate binding in the active site. This is consistent with the fact that, although EXTL3[Y670D] appears to localize normally in fibroblasts, HS levels are nevertheless reduced in affected individuals[51].

**GT47 evolution and the origin of bi-domain exostosins.** In contrast to animals, which encode only a few GT47 glycosyltransferase enzymes, plants are known to express dozens of GT47-family enzymes, which have been grouped into six clades:

GT47-A–F[17,55]. These enzymes have diverse activities and are highly important in cell wall synthesis. However, the evolutionary relationship between plant and animal GT47 enzymes has not yet been investigated. We used a Hidden Markov Model (HMM; http://bcb.unl.edu/dbCAN2/blast.php) to extract GT47-family protein sequences from diverse opisthokonts and plants. After aligning the sequences and truncating them to their GT47 domain, a maximum-likelihood phylogeny was constructed (Fig. 7). Although Arabidopsis sequences could be grouped into six distinct clades, as reported previously, we noted the existence of a seventh, well-supported clade that grouped the opisthokont sequences with sequences from the plants Ginkgo biloba and Physcomitrium patens. Although these plant enzymes do not possess a C-terminal GT64 domain, their high level of similarity with animal exostosins suggests that they could possess a related activity. We propose that this clade of sequences be named GT47-G.

Of the metazoans known to produce HS, the most distantly related to humans are the Cnidaria. However, our analysis indicated that the choanoflagellate Monosiga brevicollis (a sister to metazoans) and the poriferan sponge Amphimedon queenslandica (one of the earliest diverging metazoans) also exhibit GT47-G sequences. Interestingly, whereas the M. brevicollis sequence (Monbr1|21955) lacks a GT64 domain, A. queenslandica possesses three bi-domain exostosins (Aqu2.1.28641_001, Aqu2.1.41542_001, and Aqu2.1.32581_001) that appear orthologous to EXT1, EXT2, and EXTL3 (Fig. 7 and Supplementary Fig. 18). This suggests that the bi-domain architecture (and perhaps HS itself) arose at the outset of metazoan evolution.

**Discussion**

In this work, we used single-particle cryo-EM to produce a high-resolution map of the largest bi-domain exostosin, EXTL3. The structure reveals that EXTL3's globular domain forms a symmetrical homodimer in a very similar fashion to murine EXTL2—consistent with the prediction that its stem domain forms a homodimeric coiled coil. This apparent belt-and-braces approach suggests that dimerisation could be important for EXTL3's function. However, the increasing number of homodimeric Golgi GT structures deposited in the Protein Data Bank[56] hints that GT homodimerisation plays a more fundamental role in Golgi biology.

By conducting a phylogenetic analysis of animal and plant sequences, we established that some plants possess enzymes

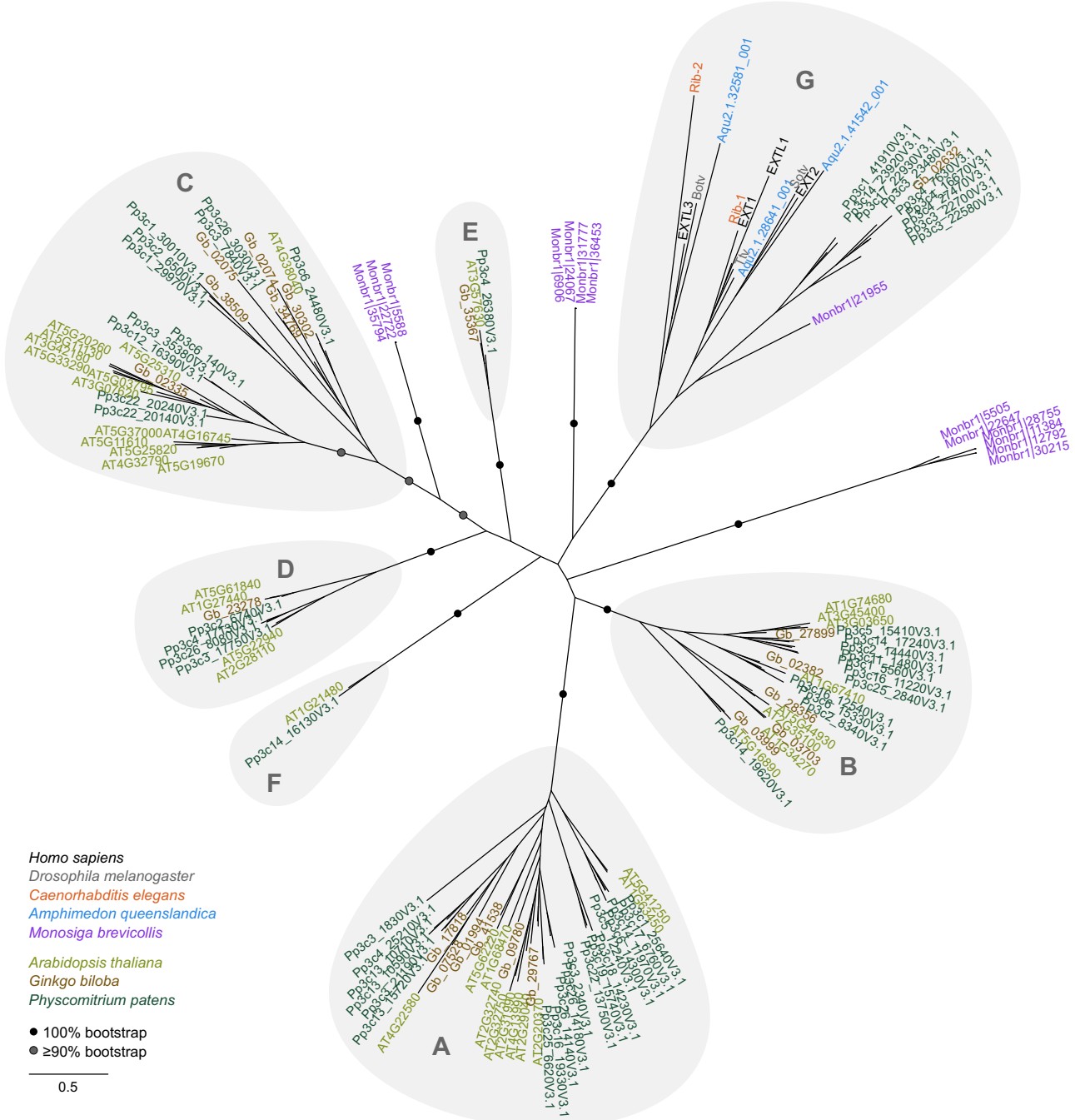

**Fig. 7 Phylogenies of GT47 domain sequences from plant, animal, and choanoflagellate proteomes indicate the existence of a seventh GT47 clade containing sequences from both plants and opisthokonts.** A phylogeny of GT47-domain sequences from *Homo sapiens*, *Drosophila melanogaster*, *Caenorhabditis elegans*, *Amphimedon queenslandica*, *Monosiga brevicollis*, *Arabidopsis thaliana*, *Ginkgo biloba*, and *Physcomitrium patens* was constructed using RAxML with 100 rapid bootstraps. Clades are labelled A–F according to reference [55]; clade G is proposed in this work. For ease of viewing, bootstraps are displayed only for major splits: black circles represent splits present in all 100 bootstrap replicates, while grey circles indicate splits present in at least 90.

closely related to exostosins. It appears that these GT47-G sequences have been omitted from several previous phylogenies of *P. patens* GT47 sequences[17,57]—perhaps due to the lack of any orthologous sequences in *Arabidopsis*. As this clade is likely to have emerged before the divergence of animals and plants, its existence calls into question the hypothesis that the ExAD-related GT47 clade (GT47-E), which contains all known GT47 sequences from chlorophyte algae, represents the origin of all plant GT47 enzymes[17,58].

Using the EXTL3 structure, we were able to determine that GT47-family glycosyltransferase domains exhibit a GT-B-type fold with a conserved Asp/Glu located on the Cα4 helix, which corresponds to a residue essential for the GlcAT-II activity of Chinese hamster EXT1[16]. Analogies with similar GT-B structures implicated this residue in donor substrate binding. Interestingly, the importance of the equivalent residue (Glu293) in GT47 member IRX10—a xylan backbone β1,4-xylosyltransferase from *Arabidopsis*[59]—has recently been demonstrated by showing that

the ectopic expression of an IRX10 E293Q mutant abrogates xylan synthesis in a dominant-negative fashion[60].

In EXTL3, however, the Cα4 helix is extended, and its conserved glutamate is engaged in a salt bridge that appears to orient the side chain so that it cannot play a catalytic role. These active-site-occluding features, as well as its apparent lack of UDP binding, indicate that the domain has likely lost activity through evolution. This is consistent with previous biochemical results[21,22]. Despite this, we were able to detect GlcAT-II activity in our preparations of EXTL3ΔN. This inverting activity, which was not fully inhibited by EDTA treatment, is unlikely to be a side-activity of the EXTL3 GT64 domain, as the GT64 catalytic mechanism is both retaining and $Mn^{2+}/Mg^{2+}$-dependent. Since the activity was substantially reduced after CRISPR-Cas9 targeting of endogenous EXT1 in the expression host, we attribute it primarily to background activity. The substantial level of background might be explained by the high sensitivity of our assay (necessary to detect the minor GlcNAcT-II activity of EXTL3) and the disproportionately high specific activity of the EXT1/2 heterodimer. It is also possible that EXTL3 further stimulates the activity of EXT1 and/or EXT2, perhaps through additional protein interactions. Nevertheless, a possibility still remains that EXTL3 possesses some intrinsic GlcAT-II activity. If so, significant conformational changes would likely be required for UDP-GlcA binding, perhaps triggered by the binding of an allosteric regulator or even the catalytic cycle of the GT64 domain.

If, as we propose, the GT47 domain has been inactivated, it raises the question as to why it has not been lost in its entirety (as it has in EXTL2). It is possible that the GT47 domain has some role in stabilising the complex—or is capable of mediating protein-protein interactions. Another explanation lies in the fact that EXTL3's stem domain (which connects the globular part to the N-terminal transmembrane helix) has been extended and rigidified by the presence of the long putative coiled coil. This suggests that physical separation of the C-terminal GT64 domain from the membrane might be important to EXTL3 function. If so, then the GT47 domain (which sits between the coiled coil and the GT64 domain) may simply contribute to the gap between membrane and GlcNAcT-I/II active site. However, some EXTL3 substrates appear to be closely associated with the membrane; for instance, glypican-1 possesses an HS addition site a mere 40 residues from its GPI anchor[61]. It is hard to explain how EXTL3's rigid stem would help it to access such a substrate. It is possible that the combined length of the stem and GT47 domain allows the GT64 domain to reach across the Golgi lumen to substrates at the opposing membrane.

Previous results indicate that bi-domain EXTL3 possesses two distinct activities: GlcNAcT-I and GlcNAcT-II activity[53]. In contrast, single-domain EXTL2 possesses GlcNAcT-I-type activity, but not GlcNAcT-II activity[62,63]. Our results confirm that, in EXTL3, both activities ought to be catalysed by the GT64 domain. Hence, EXTL2 must exhibit differences in its GT64 active site relative to EXTL3 that exclude HS backbone acceptors in favour of the (phosphorylated) linkage tetrasaccharide. In the absence of an acceptor-bound EXTL3 structure, we could not make strong conclusions about the difference in substrate specificity. Nevertheless, we note the presence of several gross structural differences between the two different acceptor binding sites, including the divergent C-termini and the presence of a nearby N-glycan (Asn790) in EXTL3. Importantly, the polybasic C-terminus appears the best explanation for EXTL3's preference for poly-acidic HS addition sites in core proteins.

In any case, we found that the EXTL3 GT64 domain active site is separated from the potential active site of the GT47 domain by a substantial distance—similarly to the two GT2 domains of E.

coli K4 chondroitin polymerase[64]. Given that the same domain organisation was predicted in the EXT1/2 heterodimer, this observation strongly suggests that the GlcAT-II and GlcNAcT-II reactions in bi-domain exostosins are not concerted. Therefore, the combination of both domains in one protein may simply help to constrain diffusion of the acceptor, thereby increasing its local concentration. Not only may this increase catalytic efficiency, but it may also protect from potential interference posed by promiscuous glycosyltransferases. This idea is consistent with the fact that Golgi GTs involved in the same pathway often form hetero-oligomeric complexes—a phenomenon that has been proposed to facilitate substrate channeling[25,65]. Hence, our structural data shed light on the requirement for fine-scale organisation of the Golgi glycosylation machinery in order to meet its demands for glycan biogenesis.

## Methods

**Expression and purification of EXTL3ΔN**. EXTL3ΔN was expressed in human embryonic kidney cells (expressing the Epstein-Barr virus nuclear antigen-1; EBNA 293) as described previously[30]. Briefly, a cDNA encoding amino acids 52–919 of human EXTL3 was cloned into expression vector pCEP4-BM40-HisTEV. EBNA 293 cells transfected with this construct were grown to confluence before undergoing a 3–4-day expression period in EX-CELL® 325 PF CHO Serum-Free Medium (Merck, Darmstadt, Germany).

Secreted EXTL3ΔN was then purified from filtered culture medium by nickel affinity chromatography (using a 1 ml HisTrap column; Cytiva, Marlborough, MA, USA) and size exclusion chromatography (using a Superdex® 200 Increase 10/300 GL column or a HiLoad 16/60 Superdex® 200 pg column; Cytiva) in a buffer containing 50 mM Tris-HCl, pH 6.8, 100 mM NaCl, and 50 mM KCl.

**CRISPR/Cas9 targeting of EXT1 in EXTL3ΔN-expressing cells**. EBNA 293 cells expressing EXTL3ΔN were transfected either with a trio of human EXT1-targeted CRISPR/Cas9 knockout plasmids (EXT1 CRISPR/Cas9 KO Plasmid (h), sc-404635; Santa Cruz Biotechnology, Dallas, TX, USA) or a non-specific CRISPR/Cas9 control plasmid (not targeting any known gene; Control CRISPR/Cas9 Plasmid, sc-418922; Santa Cruz Biotechnology) according to the manufacturer's instructions. All plasmids encoded a GFP marker to indicate transfection. The generated cells were denoted EXTL3ΔN CRISPR[EXT1] cells and EXTL3ΔN CRISPR[control] cells, respectively. Successful transfections were visually confirmed by detection of GFP via deconvolution fluorescent microscopy according to the manufacturer's instructions. The EXTL3ΔN CRISPR[EXT1] cells were then further transfected with a set of EXT1 homology-directed repair (HDR) template plasmids (EXT1 HDR Plasmid (h), sc-404635-HDR; Santa Cruz Biotechnology) for permanent expression according to the manufacturer's instructions. Cells that had undergone HDR were then selected by virtue of their puromycin resistance (derived from the integrated HDR template).

**Deconvolution immunofluorescence microscopy**. Prior to EXT1 HDR plasmid transfection, expression of EXT1 in EXTL3ΔN CRISPR[EXT1] cells and EXTL3ΔN CRISPR[control] cells was examined by immunofluorescence microscopy[66]. In detail: cells were washed with PBS (137 mM NaCl, 2.7 mM KCl, 8 mM $Na_2HPO_4$, and 2 mM $KH_2PO_4$, pH 7.4) and fixed in acetone in order to retain cellular and sub-cellular structures. The fixed cells were first pre-coated with 10% anti-mouse total Ig and then exposed to primary anti-EXT1 antibody (A-7, sc-515144; Santa Cruz Biotechnology; dilution 1:100) overnight. After extensive washings with PBS, the cells were treated with Alexa Fluor 594-tagged goat anti-mouse IgG (A-11005; ThermoFisher, Waltham, MA, USA; dilution 1:500) for 4 h. To visualise nuclei, DNA staining was performed with 4′,6-diamidino-2-phenylindole (DAPI; ThermoFisher; diluted to 300 μM), as well as staining with antibodies, as recommended by the manufacturers. In the controls, the primary antibody was omitted. The fluorescent images were analysed by using a Carl Zeiss AxioObserver inverted fluorescence microscope with deconvolution technique and equipped with objective EC "Plan-Neofluar" 63 X/1.25 Oil M27 and AxioCam MRm Rev Camera. Identical exposure settings and times were used for all images. During microscopy, the entire slides were scanned and immunofluorescence images at 20 X and 100 X magnifications were captured. The low magnification images were used to identify representative locations for high magnification images. Quantification of EXT1 expression level was analysed by scanning entire 20 X magnification images using Zeiss AxioVision Release 4.8 software.

**Slot blot**. Cells ($10^4$ cells) were extracted with radio-immunoprecipitation assay (RIPA) buffer (0.1% w/v SDS, 0.5% v/v Triton X-100, 0.5% w/v sodium deoxycholate in PBS supplemented with proteinase inhibitors (cOmplete mini) and 0.5 mM phenylmethylsulfonyl fluoride) at 4 °C. RIPA extracts were analysed by slot blotting to PVDF membranes that were incubated with anti-EXT1 antibody (A-7, sc-515144; 1:200 dilution) or anti-actin beta antibody (AC-15 anti-β-actin-

peroxidase, A3854; Sigma-Aldrich, St Louis, MO, USA; 1:25,000 dilution) followed by visualization using horseradish peroxidase-conjugated anti-mouse IgG (Bio-Rad, Hercules, CA, United States; 172-1011; dilution 1:500). The membranes were developed by chemiluminescence (GE Healthcare, Sweden) using a Fujifilm detector. In the controls, the primary antibody was omitted. Beta-actin was used as loading control. Staining intensities were recorded by densitometry using Gel-Pro Analyser software, version 3.0.00.00. Images from uncropped blots (as well as all other uncropped gels) are provided in Supplementary Fig. 19.

**Peptide mass spectrometry.** Protein solutions were reduced (DTT) and alkylated (iodoacetamide) and subjected to enzymatic digestion with sequencing-grade trypsin (Promega, Madison, WI, USA) overnight at 37 °C. After digestion, the supernatant was pipetted into a sample vial and loaded onto an autosampler for automated LC-MS/MS analysis. All LC-MS/MS experiments were performed using a Dionex Ultimate 3000 RSLC nanoUPLC (Thermo Fisher Scientific Inc, Waltham, MA, USA) system and a Q Exactive Orbitrap mass spectrometer (Thermo Fisher Scientific Inc, Waltham, MA, USA). Separation of peptides was performed by reverse-phase chromatography at a flow rate of 300 nl min$^{-1}$ and a Thermo Scientific reverse-phase nano Easy-spray column (Thermo Scientific PepMap C18, 2 μm particle size, 100 Å pore size, 75 μm i.d. × 50 cm length). Peptides were loaded onto a pre-column (Thermo Scientific PepMap 100 C18, 5 μm particle size, 100 Å pore size, 300 μm i.d. × 5 mm length) from the Ultimate 3000 autosampler with 0.1% formic acid for 3 min at a flow rate of 10 μl min$^{-1}$. After this period, the column valve was switched to allow elution of peptides from the pre-column onto the analytical column. Solvent A was 0.1% formic acid and solvent B was 80% acetonitrile with 0.1% formic acid. The linear gradient employed was 2–40% B in 30 min. Further wash and equilibration steps gave a total run time of 60 min.

The LC eluant was sprayed into the mass spectrometer by means of an Easy-Spray source (Thermo Fisher Scientific Inc.). All $m/z$ values of eluting ions were measured in an Orbitrap mass analyzer, set at a resolution of 35,000 and scanning range between $m/z$ 380–1500. Data-dependent scans (Top 20) were employed to automatically isolate and generate fragment ions by higher energy collisional dissociation (HCD, NCE:25%) in the HCD collision cell and measurement of the resulting fragment ions was performed in the Orbitrap analyser, set at a resolution of 17,500. Singly charged ions and ions with unassigned charge states were excluded from MS/MS and a dynamic exclusion window of 20 s was employed.

Post-run, the data was processed using Protein Discoverer (version 2.3, ThermoFisher). Briefly, all MS/MS data were submitted to the Mascot search algorithm (Matrix Science, London UK, version 2.6.0) and searched against a common contaminants database (cRAP 20190401, 125 sequences; 41129 residues) and the UniProt human database (CCP_UniProt_homo_sapiens_proteome_20180409 database, 93,609 sequences; 37,041,084 residues). Variable modifications of oxidation (M), deamidation (NQ), and a fixed modification of carbamidomethyl (C) were applied. Peak areas for each identified peptide were generated and combined to give protein abundance. The peptide and fragment mass tolerances were set to 25 ppm and 0.1 Da, respectively.

Scaffold (version Scaffold_4.10.0, Proteome Software Inc., Portland, OR) was used to validate MS/MS based peptide and protein identifications. Peptide identifications were accepted if they could be established at greater than 90.0% probability by the Scaffold Local FDR algorithm. Protein identifications were accepted if they could be established at greater than 99.0% probability and contained at least two identified peptides. Protein probabilities were assigned by the Protein Prophet algorithm[67]. Proteins that contained similar peptides and could not be differentiated based on MS/MS analysis alone were grouped to satisfy the principles of parsimony. Proteins sharing significant peptide evidence were grouped into clusters.

**Preparation of K5 heparosan oligosaccharide acceptors.** DP10 lyase product from *E. coli* K5 capsular polysaccharide was purchased from Elicityl (Crolles, France). To remove the non-reducing-terminal Δ-4,5-unsaturated uronic acid, the DP10 oligosaccharide was treated with BT4658$^{GH88}$ from *Bacteroides thetaiotamicron* VPI-5482[68] in an overnight reaction containing 6 mM DP10 oligosaccharide, 50 mM ammonium acetate, pH 5.5, and 0.3 mg ml$^{-1}$ BT4658$^{GH88}$ at 30 °C. BT4658GH88 was a gift of Cleton Santos (LNBr/CNPEM, Brazil). Glycosidase was removed by passing the sample through a 30 kDa NanoSep centrifugal filter (Pall, New York, USA). The resultant DP9 GlcNAc-[GlcA-GlcNAc]$_4$ oligosaccharide was in turn converted to DP8 [GlcA-GlcNAc]$_4$ by treatment with *Pa*GH89 α-*N*-acetylglucosaminidase (Novozymes, Bagsværd, Denmark) in an overnight reaction containing 6 mM DP9 oligosaccharide, 25 mM ammonium acetate, pH 5.5, and 0.4 mg ml$^{-1}$ *Pa*GH89 at 30 °C. As before, glycosidase was removed by passing the sample through a 30 kDa NanoSep centrifugal filter.

**K5 heparosan extension assay.** Reactions were carried out in a thermocycler at 37 °C for 15 min, 1 h, or 16 h before termination at 99 °C for 10 min. Each 10 μl reaction contained 25 mM ammonium acetate, pH 6.5, 10 μM DP8/DP9 heparosan acceptor, 25 μg ml$^{-1}$ purified EXTL3ΔN, 1.5 mM UDP-GlcNAc/UDP-GlcA, 3 mM MnCl$_2$, and 3 mM MgCl$_2$. For quantitative assays, protein concentration was adjusted by BCA assay for every technical replicate. For cofactor depletion experiments, MnCl$_2$ and MgCl$_2$ were replaced with 10 mM EDTA. Completed

reactions were dried using a centrifugal evaporator. For β-glucuronidase treatment, products were resuspended in 10 μl 50 mM ammonium acetate, pH 5.5, supplemented with either 0.1 mg ml$^{-1}$ bovine liver β-glucuronidase, type B-1 (*Bt*GUSB; Merck) or 50 μg ml$^{-1}$ *Thar*GH79a from *Trichoderma harzianum* (THAR02_03122; GenBank: KKP04785.1; Novozymes) before incubating at 37 °C overnight and re-drying. Derivatisation of reducing ends with 8-aminonaphthalene-1,3,6-trisulfonic acid (ANTS) was then achieved by resuspending each sample in 15 μl of labelling reagent containing 33 mM ANTS, 0.45 M 2-picoline-borane, 67% DMSO, and 5% acetic acid and incubating at 37 °C overnight. Derivatised products were then analysed by Polysaccharide Analysis using Carbohydrate Electrophoresis (PACE) as described previously[69]. Briefly, samples were dried using a GeneVac miVac DNA centrifugal concentrator (Genevac Ltd, Ipswich, UK) at 60 °C before being resuspended in 5 μl 6 M urea. From each sample, 2.5 μl was loaded into a 240 × 180 × 0.75 mm polyacrylamide gel comprising a stacking gel with 10% polyacrylamide and a resolving gel with 20% polyacrylamide, both containing 0.1 M Tris-borate, pH 8.2. Gels were run in 0.1 M Tris-borate buffer in a Hoefer SE660 electrophoresis tank (Hoefer Inc, Holliston, MA, USA) at 200 V, 5 mA for 30 min and then 1000 V, 30 mA for 135 min before imaging using a G-box (Syngene, Cambridge, UK) fitted with 365 nm UV tubes, applying a 500–600 nm short pass detection filter. Images were acquired with GeneSnap software (Syngene, version 7.12).

**Oligosaccharide mass spectrometry.** Reactions were carried out as above, except that volumes were scaled up by a factor of ten. Completed reactions were passed through a 10 kDa NanoSep centrifugal filter (Pall) and dried using a centrifugal concentrator before desalting with a Dowex (50WX8, Na$^+$ form, 100–200 mesh; Merck) cation exchange column as described previously[70]. Briefly, Dowex beads were washed thrice in 4 M HCl before extensive washing in MilliQ water followed by three washes with 5% acetic acid. A glass Pasteur pipette was plugged with glass wool and fitted with a valve (consisting of a piece of hose closed with a screw compressor clamp) before packing with 500 μl Dowex beads. The beads were then washed with 1 ml 5% acetic acid; flow was permitted until the top of the liquid phase drew level with the top of the resin bed. Dried sample was resuspended in 50 μl 5% acetic acid and loaded onto the column; just enough liquid was allowed to drain so that the meniscus returned to the previous level. After placing the hose in a collection vial, the sample was then eluted with further 5% acetic acid. The eluate was dried in a centrifugal evaporator before derivatisation using 2-aminobenzamide (2-AB), followed by GlycoClean S (Agilent, Santa Clara, CA, USA) clean-up, as described previously[70]. Briefly, dried sample was resuspended in 10 μl labelling reagent (60 mg ml$^{-1}$ 2-AB and 60 mg ml$^{-1}$ sodium cyanoborohydride in 3:7 acetic acid: DMSO) and incubated at 65 °C for 3 h. Each GlycoClean S cartridge was washed with 1 ml MilliQ water, then 5 ml 30% acetic acid, then 4 ml 100% acetonitrile. After cooling to room temperature, labelled sample was spotted onto the wetted cartridge membrane and incubated for 15 min. The cartridge was washed with 1 ml 100% acetonitrile followed by five sequential 1 ml washes with 96% acetonitrile. The labelled oligosaccharides were then eluted with 1.5 ml MilliQ water. After drying, samples were resuspended in water and mixed 1:1 (v/v) with 2,5-dihydroxybenzoic acid (DHB) matrix (20 mg ml$^{-1}$ in 50% methanol with 0.4 mg ml$^{-1}$ ammonium sulfate) before spotting 2 μl on a ground-steel target plate for mass spectrometry. Data were collected on an ultrafleXtreme MALDI/TOF-TOF instrument (Bruker) using a 2-kHz smartbeam-II laser and acquired on reflector negative ion mode (mass range 1000–4500 Da). On average, 10,000 shots were used to obtain high enough resolution. Bruker flexControl and flexAnalysis software (versions 3.4 and 3.4, respectively) were used for acquisition and analysis, respectively.

**Cryo-EM sample preparation.** Purified EXTL3ΔN was concentrated to 0.1–0.4 mg ml$^{-1}$. Using a PELCO easiGlow system (Ted Pella), QUANTIFOIL® holey carbon grids (R 1.2/1.3 for apo-structure; R 2/2 for UDP-bound structure) were glow-discharged for 60 s at 25 mA, before application and vitrification of a 3 μl protein sample using a FEI Vitrobot Mark IV system (Thermo Fisher Scientific) at 4 °C and 95% humidity, with a blot time of 3 s. For the UDP-bound structure, UDP and MnCl$_2$ were added to the protein preparation 150 min before grid freezing at final concentrations of 10 mM and 2.5 mM, respectively.

**Cryo-EM data collection.** Grids were screened using a 200 kV Talos Arctica microscope (Thermo Fisher Scientific) and movies were collected using a 300 kV Titan Krios microscope (Thermo Fisher Scientific) at the BiocEM facility, Department of Biochemistry, University of Cambridge. All data collection parameters are listed in Supplementary Table 7.

**Cryo-EM data processing.** Refinement parameters are listed in Supplementary Table 5. Apo-EXTL3 was processed in RELION 3.0.1[71]. Initially, 1440 movies were corrected for beam-induced motion using MotionCor2[72]. CTF estimation was performed using Gctf v1.06[73]. A total of 656,292 particles were extracted using autopicking (binned 3×). After several rounds of 2D classification, an initial model with C1 symmetry was constructed using an SGD algorithm implemented in RELION. Several rounds of 3D classification reduced the total number of particles to 171,285. These particles were re-extracted with a 420 px box size (binned 1×); subsequently, a

new initial model with C2 symmetry was constructed, and a 3D refinement was performed. After movie refinement and particle polishing, 3D refinement and post-processing at 0.67 Å/px were used to create the final map with 2.4 Å resolution (FSC = 0.143 criterion). Ab initio model-building was achieved using Buccaneer[74] and Coot[75] (version 0.8.9.2) via the CCP-EM interface[76] (version 1.4.1) before refinement in ISOLDE[77] (on the UCSF ChimeraX[78] platform, version 1.1) and Phenix real-space refine[79] (Phenix version 1.18.2-3874). Refinement parameters are listed in Supplementary Table 7.

The UDP-bound structure was processed using RELION 3.0.1[71]. Initially, 2573 movies were corrected for beam-induced motion using MotionCor2. The CTF was estimated using Gctf v1.06. After visual inspection, 38 movies were removed before auto-picking and extraction of 1,133,069 particles with a box size of 240 Å (binned 3×). These particles underwent several rounds of 2D classification before construction of an initial 3D model via an SGD algorithm implemented in RELION. A single round of 3D classification produced a 3D reference to auto-pick and extract 1,696,344 particles afresh (binned 3×). The same reference was used for further 3D classification in order to select 238,379 particles that were subsequently re-extracted with a box size of 360 px (binned 1×). After finding no discernible differences between the two monomers, a new initial model with C2 symmetry was constructed in order to conduct a further 3D refinement. Two rounds of CTF refinement and Bayesian polishing were carried out before post-processing at 1.06 Å/px to create the final map with 2.9 Å resolution (FSC = 0.143 criterion). The apo-structure was aligned to the map and used as a starting model before further refinement in Coot, ISOLDE, and Phenix real-space refine. Refinement parameters are listed in Supplementary Table 7.

**Sequence alignments, logos, and phylogenetics**. For the sequence alignment of characterised GT47s, sequences were downloaded manually from UniProtKB (https://www.uniprot.org/) or from TAIR (https://www.arabidopsis.org/). The C-terminal halves of the human protein sequences were removed before alignment of all sequences with MUSCLE[80,81] v3.8.31.

To extract GT47 domain-containing sequences from further animal and plant proteomes, proteome models were downloaded for *Homo sapiens* (from NCBI Genome; https://www.ncbi.nlm.nih.gov/genome/), *Drosophila melanogaster* (NCBI), *Caenorhabditis elegans* (NCBI), *Amphimedon queenslandica* (EnsemblMetazoa; https://metazoa.ensembl.org/Amphimedon_queenslandica/Info/Index), *Monosiga brevicollis* MX1[82] (JGI MycoCosm; https://mycocosm.jgi.doe.gov/Monbr1/Monbr1.home.html), *Arabidopsis thaliana* (JGI Phytozome v12; https://phytozome.jgi.doe.gov/pz/portal.html), *Ginkgo biloba*[83,84] (GigaDB; http://gigadb.org/dataset/100613), and *Physcomitrium patens* (JGI Phytozome v12). A GT47 Hidden Markov Model (HMM) was downloaded from the dbCAN2 server[85,86] (http://bcb.unl.edu/dbCAN2/) and used to search the proteomes for GT47-family sequences using hmmsearch from the HMMER package[87](version 3.1b2) with an *E*-value cut-off of $10^{-10}$. For the animal sequences, the GT64 domain portion was removed manually for each sequence following a preliminary alignment of these sequences with MUSCLE. Similarly, the sulfatase domain of Monbr1|21955 was removed. All sequences were then aligned with MAFFT v7.310, and each aligned sequence was truncated to a portion corresponding to residues 196–538 of human EXTL3 using a custom python (version 2.7.17, alitrunc.py) script. ProtTest 3[88,89] (version 3.4.2) was subsequently used to determine the most appropriate model of protein evolution, before constructing a phylogeny using RAxML[90,91] (version 8.2.11) with 100 rapid bootstraps. The resultant tree was visualised using FigTree v1.4.4 and edited for publication using Inkscape (version 1.0.1).

**Figure preparation**. All figures were prepared using Inkscape. Structures were rendered using ChimeraX and PyMOL (version 2.4.0a0 Open-Source).

**Reporting summary**. Further information on research design is available in the Nature Research Reporting Summary linked to this article.

## Data availability
The data that support this study are available from the corresponding authors upon reasonable request. Numerical data for Fig. 2d,e, Supplementary Fig. 2e, and Supplementary Fig. 3b can be found in the Source Data file. Cryo-EM maps for apo- and UDP-bound EXTL3 were deposited in the Electron Microscopy Data Bank under accession codes EMD-11923 (apo structure) and EMD-11926 (UDP-bound structure). Atomic co-ordinates for apo- and UDP-bound EXTL3 were submitted to the Protein Data Bank under accession codes 7AU2 (apo structure) and 7AUA (UDP-bound structure). The mass spectrometry proteomics data have been deposited to the ProteomeXchange Consortium via the PRIDE partner repository with the dataset identifiers PXD032145 (regular EXTL3ΔN purification) and PXD032144 (CRISPR experiments). Proteomic search databases are available at UniProt (human proteome reference [https://www.uniprot.org/proteomes/UP000005640]) and The Global Proteome Machine (cRAP common contaminants database [https://www.thegpm.org/crap/]). Individual protein sequences were downloaded from the UniProtKB (*Hs*EXT1: Q16394; *Hs*EXT2: Q93063; *Hs*EXTL1: Q92935; *Hs*EXTL3: O43909) or TAIR (*At*MUR3: AT2G20370; *At*XLT2: AT5G62220; *At*XUT1: AT1G63450; *At*ARAD1: AT2G35100; *At*IRX10: AT1G27440; *At*IRX10L: AT5G61840; *At*IRX7: AT2G28110;

*At*IRX7L: AT5G22940; *At*ExAD: AT3G57630; *At*XGD1: AT5G33290) databases. AlphaFold pre-computed structural predictions are available from the AlphaFold Protein Structure Database (*Hs*EXT1: Q16394; *Hs*EXT2: Q93063; *Hs*EXTL3: O43909). Proteome models are available at NCBI Genome (*Homo sapiens*: RefSeq GCF_000001405.40; *Drosophila melanogaster*: RefSeq GCF_000001215.4; *Caenorhabditis elegans*: RefSeq GCF_000002985.6), JGI Phytozome (*Arabidopsis thaliana*: 167 [https://phytozome-next.jgi.doe.gov/info/Athaliana_TAIR10]; *Physcomitrium patens*: 318 [https://phytozome-next.jgi.doe.gov/info/Ppatens_v3_3]), JGI MycoCosm (*Monosiga brevicollis*: Monosiga brevicollis MX1 [https://mycocosm.jgi.doe.gov/Monbr1/Monbr1.home.html]), EnsemblMetazoa (*Amphimedon queenslandica*: Aqu1 [https://metazoa.ensembl.org/Amphimedon_queenslandica/Info/Index]), and GigaDB (*Ginkgo biloba*: 100613). The GT47 Hidden Markov Model used in this work is available from the dbCAN2 server (dbCAN-HMMdb-V9). Source data are provided with this paper.

## Code availability
The short Python script used to truncate alignments in this work (alitrunc.py) is available at Zenodo (https://doi.org/10.5281/zenodo.6562402). Graphical views of phylogenetic trees were made using FigTree v1.4.4. [http://tree.bio.ed.ac.uk/software/figtree/].

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

## Acknowledgements
This work was funded by grants from the University of Cambridge, OpenPlant (BB/L014130/1, P.D.), the Swedish Research Council (2014-03402, K.M.; 2016-04855, D.T.L.), Cancerfonden (21 1426 Pj 01 H, K.M.), and the Wellcome Trust (200873/Z/16/Z, B.F.L.). L.F.L.W. was supported by the University of Cambridge. T.D. was supported by an AstraZeneca studentship. We thank Professor emeritus Ingemar Carlstedt for financial support to the Glycobiology group. We thank the Lund Protein Production Platform, Lund University, Sweden (http://www.lu.se/lp3) for providing support for protein purification and the Cambridge Centre for Proteomics, Cambridge University, UK, for carrying out proteomic analyses. We thank Lee Cooper, University of Cambridge, for help with grid preparation and Clelton Santos, State University of Campinas, for glycosidase preparation.

## Author contributions
L.F.L.W., P.D., K.M., and D.T.L. designed experiments. K.M. performed protein expressions and CRISPR-Cas9 experiments. S.W.H., L.F.L.W., and T.D. performed protein purification and cryo-EM sample preparation. L.F.L.W. and A.E.-P. carried out activity assays. T.T. and L.F.L.W. carried out carbohydrate mass spectrometry. K.B.R.M.K. identified a relevant glycosyl hydrolase. D.Y.C. performed cryo-EM data collection. T.D. and L.F.L.W. processed the cryo-EM data. L.F.L.W. wrote the manuscript. P.D., D.T.L., K.M., and B.F.L. supervised the project and contributed to the manuscript.

## Competing interests
The authors declare the following competing interest: KBK is an employee of Novozymes, which is a major enzyme producing company. The remaining authors declare no competing interests.
