## [Peer Review File · Nature Communications]

The structure of EXTL3 helps to explain the different roles of bi-domain exostosins in heparan sulfate synthesisReviewers' Comments:

Reviewer #1:

Remarks to the Author:

Comments on "The structure of EXTL3 explains the different contributions of bidomain exostosins to heparan sulfate synthesis" by Wilson et al.

Remarks to the author

The study by Wilson et al. describes the cryo-EM structures of the exostosin EXTL3, both in apo and UDP-bound states. The authors performed expression and purification of a truncated version of EXTL3, lacking the N-terminal transmembrane helix, and elucidated EXTL3 structures using single-particle cryo-EM. Based on this structural information, the authors described the architecture of the two glycosyltransferase (GT) domains, GT64 and GT47, and explain why GT47 is inactive in EXTL3. However, this work is very speculative, and it is not clear the conceptual advance it offers. Even though the authors described an in-vitro activity assay for the GlcNAc-TII activity of EXTL3, they did not use it to validate any of the structural observations that were described in the manuscript. Therefore, the paper dwells around a redundant description of the architecture of the proteins without offering biochemical and functional insights into the protein function.

Main Points

- The authors provide neither functional nor structural data that would explain the alternating saccharide pattern in heparan sulfate, as they claimed in the abstract.
- The manuscript is repetitive and hard to read. Some sections do not provide any relevant information and should be shortened, and the corresponding figures moved to the supplementary information. For example, the section on glycans attached to EXTL3. The visualization of glycans attached to glycoproteins in cryo-EM maps is well known and has been described multiple times in different cryo-EM structures of eukaryotic glycoproteins. Therefore, it is unnecessary to have a whole section (plus a main figure) describing the attached glycans observed in EXTL3, especially when there is no insight into the function of these glycans in EXTL3 activity.
- The discussion regarding the C-terminal extension of EXTL3 compared to EXT2 is very speculative. There is no evidence at all of this region having any role in the catalytic activity of EXTL3. The high sequence conservation observed does not mean that it has an important function in EXTL3 activity, as the authors claimed. Again, it is only a description of a structural feature without any biochemical or functional support.
- The structure of EXTL3 with UDP bound allows the visualization of several residues involved in UDP binding. However, the authors missed the opportunity of validating these observations using their in-vitro activity assay with different EXTL3 mutants.
- Many glycosyltransferases that use nucleotide-activated sugars as donor substrates can hydrolyze them in the absence of an acceptor substrate. If this is why only UDP was observed in their structure, it should be mentioned in the text. If not, it is not clear why the authors used UDP instead of UDP-GlcNAc for the structural studies, which would provide more information regarding the sugar specificity of the GT64 domain.
- The explanation of why the GT47 domain is inactive lacks biochemical validation. The authors should demonstrate that this domain cannot bind donor substrate in contrast to active GT47 domains of other proteins.
- The discussion section is mostly a repetition of the results. It lacks a discussion of what their results offer in the context of EXTL3 being a membrane protein. How would the GT domains be oriented relative to the membrane?

Minor points:

- Mainly being a structural work, the authors should provide the pipeline of the cryo-EM data processing.
- The authors should state that there are five different exostosin genes annotated in mammals and

described why they focused on EXTL3 so that it is clear for a broader readership.

- Panel b in Fig2 is repeated in Fig.3 panel b. This view of the enzyme is constantly repeated in several figures.
- Fig 2: N and C-termini of the two chains in the dimer should be labeled for better understanding. Gray coloring of the interdomain region cannot be easily visualized, and it should be changed to improve the contrast.
- Fig 4: The amino acids in the protein sequence alignment are not numbered.
- There is a typo in the figure legend of Fig 6. DDD needs to be changed to DxD
- The formatting of tables should be consistent in the supplementary file.
- Figure legends need to be self-explanatory. Many figure legends are only titles that do not supply enough information.

Reviewer #2:

Remarks to the Author:

Heparan sulfate (HS) is a glycosaminoglycan (GAG) polysaccharide of alternating α -1,4-linked N-acetylglucosamine (α -GlcNAc) and β 1,4-linked glucuronic acid (β -GlcA). The HS backbone synthesis is catalyzed by enzymes belonging to the exostosin (EXT) family. These enzymes generally have two glycosyltransferase (GT) domains. The N-terminal domain (classified as CaZy family GT47) is thought to catalyze the β -GlcA transfer, whereas the C-terminal domain (classified as GT64) is involved in addition of α -GlcNAc residue. However, detailed knowledge of the dual functional abilities of EXT family members are limited, mainly due to lack of their structural evidence.

In this manuscript, Wilson et al. reported high-resolution cyto-EM structures of soluble form of an EXTL3 homodimer in the apo-form and bound with UDP and Mn²⁺. Based on the obtained EXTL3 structures, the authors provided structural characteristics of the respective GT domains and their functional importance in EXT-mediated HS backbone synthesis. The authors also proposed that EXTs achieve ordered monosaccharide addition by a simple dissociative mechanism. Experiments are carefully performed and all data presented in this manuscript is convincing. Therefore, this reviewer feels that the manuscript is acceptable in principle, but still requires some revisions.

Major points #1

The authors described that several unique features observed in EXTL3 GT47 domain could be the leading causes for its loss of enzymatic activity to transfer GlcA units. Additional experimental data to support this notion should be required. For example, does genetic engineering of EXT1 to have substitution/insertion of characteristic amino acid residues observed in EXTL3 affect enzymatic activity of the recombinant EXT1 protein? Comparison of the modeled structure of EXT1 with that of EXTL3 would also help understand the proposed molecular mechanisms of GT47-mediated enzymatic activities of EXT proteins.

Major points #2

The authors estimated the distance between the GT47 and GT64 active sites in the EXTL3 structure. Based on the results shown in Fig. 6, they speculated the modes of action of EXT family enzymes. Additional figures showing the domain organization of proposed (or simulated) EXT1 structure would strengthen this interpretation.

Minor points #1

Page 5, line 20, To clarify the importance of a pair of intermolecular disulfide bridges between Cys793 and Cys951 in the EXTL3 homodimer formation, this disulfide structure should be shown in the figure.

Minor points #2

Page 8, lines 9 and 16, citations of Fig. 5d and Fig. 5e in the text should be replaced with Fig. 4d and Fig. 4e, respectively.

Minor points #3

Page 8, line 17, the authors reported N-glycosylated sites of EXTL3 in detail. Their functional relevance in EXTs should also be discussed.

Hiroshi Kitagawa

REVIEWER COMMENTS

Reviewer #1 (Remarks to the Author):

Comments on “The structure of EXTL3 explains the different contributions of bidomain exostosins to heparan sulfate synthesis” by Wilson et al.

Remarks to the author

The study by Wilson et al. describes the cryo-EM structures of the exostosin EXTL3, both in apo and UDP-bound states. The authors performed expression and purification of a truncated version of EXTL3, lacking the N-terminal transmembrane helix, and elucidated EXTL3 structures using single-particle cryo-EM. Based on this structural information, the authors described the architecture of the two glycosyltransferase (GT) domains, GT64 and GT47, and explain why GT47 is inactive in EXTL3. However, this work is very speculative, and it is not clear the conceptual advance it offers. Even though the authors described an in-vitro activity assay for the GlcNAc-TII activity of EXTL3, they did not use it to validate any of the structural observations that were described in the manuscript. Therefore, the paper dwells around a redundant description of the architecture of the proteins without offering biochemical and functional insights into the protein function.

We thank the reviewer for these comments. We respectfully disagree that the work is very speculative, especially as we use the structural data to explain previously reported biochemical data. As we describe in the manuscript, the EXTL3 structure provides reasonable explanations for the differences between EXTL3 and EXT1 activity, the role of certain exostosin mutations in disease, and the biochemistry and evolution of GT47 family members at large; hence, we do not believe that the structural information is redundant. However, most importantly, key to our structural findings is not just the architecture of the individual protein domains, but the architecture of the protein as a whole. Although GT-GT fusions and complexes appear to be common in the eukaryotic Golgi, to our knowledge, this is the first structure of such an enzyme, and is an important advance in understanding why such interactions have evolved. Our data help to rule out the existence of an elegant mechanism of substrate channelling in such cases.

Nevertheless, we appreciate that further biochemical experiments would have complemented our results. Consequently, we have included further experiments (Fig. 1c, Fig. 2, Supplementary Figs. 1–3, and Supplementary Table 3), data analysis, and manuscript modifications that we hope will satisfy the reviewer’s concerns. Please see our responses to the main points below. The new biochemical work, which involved CRISPR/Cas9 targeting of EXT1 in the expression host, was technically challenging and we believe that it represents an achievement in itself.

Out of respect for the reviewer’s comments, we also attempted to express and purify four EXT1 variants (with and without various mutations) by two different approaches, with the aim of introducing EXTL3-like features into the active site of EXT1’s GT47 domain. Unfortunately, expression of EXT1 was not straightforward and we were unable to produce protein for biochemical assays. Nevertheless, our new analysis of exostosin AlphaFold

models (Supplementary Fig. 9) supports our existing hypothesis that the nucleotide sugar binding site is not occluded in EXT1 as it is in EXTL3. Additionally, a new AlphaFold-Multimer model of the EXT1/EXT2 heterodimer (Supplementary Fig. 16) has strengthened our conclusion regarding the general mechanism of HS backbone extension. Finally, an observation on the evolutionary history of bi-domain exostosins has been added to consolidate the impact of the phylogenetic work (Supplementary Fig. 18).

When combined, we believe that the new data and analyses have substantially improved our manuscript. It is our earnest hope that readers will recognise the advances made in understanding the organisation of Golgi glycosylation and the evolution of glycosyltransferases.

Main Points

- The authors provide neither functional nor structural data that would explain the alternating saccharide pattern in heparan sulfate, as they claimed in the abstract.

This statement has now been removed from the abstract. It was not a main finding of the paper. We conclude from our structural studies that the alternating structure is likely to be a simple dissociative mechanism, since the active sites are not arranged in a way that can obviously channel the acceptor.

- The manuscript is repetitive and hard to read. Some sections do not provide any relevant information and should be shortened, and the corresponding figures moved to the supplementary information. For example, the section on glycans attached to EXTL3. The visualization of glycans attached to glycoproteins in cryo-EM maps is well known and has been described multiple times in different cryo-EM structures of eukaryotic glycoproteins. Therefore, it is unnecessary to have a whole section (plus a main figure) describing the attached glycans observed in EXTL3, especially when there is no insight into the function of these glycans in EXTL3 activity.

In response to this comment, the manuscript has undergone substantial revision to reduce repetition and improve readability. In addition:

- The sections on the energetics of homodimerisation (lines 192–195) has been significantly reduced and moved in part to the supplementary information (Supplementary Table 4)
- The section on N-glycosylation (lines 184–185) has been significantly reduced and moved in part to the supplementary information (please see Supplementary Fig. 7)
- A comment has been added to the discussion regarding the proximity of Asn790 to the potential acceptor binding site (please see Fig. 4c and line 406).
- The section on disease mutations in EXT1 and EXT2 has been removed
- The section on Chinese hamster EXT1 point mutants has been redistributed (lines 255–258).

- The discussion regarding the C-terminal extension of EXTL3 compared to EXT2 is very speculative. There is no evidence at all of this region having any role in the catalytic activity of EXTL3. The high sequence conservation observed does not mean that it has an important function in EXTL3 activity, as the authors claimed. Again, it is only a description of a structural feature without any biochemical or functional support.

We agree that our interpretations here are speculative, and regret that we did not have the opportunity to test them experimentally. This section has been shortened and the sequence

alignment has been removed. However, we maintain that our interpretations are reasonable and not fanciful. The C-terminal extension is proximal to the putative acceptor binding site, whose location can be predicted by comparison with EXTL2 (this is now stated clearly in lines 205–207 and Fig. 4c). Therefore, it is plausible that it could interact with parts of the acceptor (especially the underlying core protein of the proteoglycan), and thus could be relevant to the GlcNAcT-II efficiency. We did not find any other similarly electropositive regions on the protein surface that could explain the well-known preference for polyacidic core protein sites. To our knowledge, this is the only reasonable explanation that has been offered for the phenomenon, and we believe it is worthy of mention.

- The structure of EXTL3 with UDP bound allows the visualization of several residues involved in UDP binding. However, the authors missed the opportunity of validating these observations using their *in-vitro* activity assay with different EXTL3 mutants.

Unfortunately, we do not have the resources to make a range of EXTL3 point mutants. Nevertheless, we were able to refer to previous *in vivo* work on the Y670D mutation to support the importance of these residues (lines 314–318). Furthermore, we do not expect that the reported GT64 UDP binding site will be controversial, as it is extremely similar to that seen in several EXTL2 structures, and so believe additional experimental support is not warranted.

- Many glycosyltransferases that use nucleotide-activated sugars as donor substrates can hydrolyze them in the absence of an acceptor substrate. If this is why only UDP was observed in their structure, it should be mentioned in the text. If not, it is not clear why the authors used UDP instead of UDP-GlcNAc for the structural studies, which would provide more information regarding the sugar specificity of the GT64 domain.

We apologise that this was not made clearer in the manuscript; we have undertaken changes to better explain this (lines 264–272). Free UDP is routinely added to glycosyltransferases during structure determination (to form an enzyme–product complex) in order to provide insight into generic ligand binding. The structure of EXTL2 bound to UDP-GlcNAc has already been solved by crystallography (Pedersen et al., 2003 doi: 10.1074/jbc.M210532200), hence we did not see much added value in obtaining a complex of UDP-GlcNAc with EXTL3, which exhibits almost identical residues in the donor substrate binding site. Furthermore, in this particular case, UDP should be able to bind not only to UDP-GlcNAc binding sites, but also to any possible UDP-GlcA binding sites, potentially capturing all active sites in one structure. This is why we added free UDP, and not UDP-GlcNAc, to our protein prior to grid freezing (as described in the materials and methods). As a result, we were able to observe from the structural data that UDP does not appear to bind to the GT47 domain (Supplementary Fig. 15), even when present at a concentration of 10 mM (about 100–1,000-fold greater than the K_d for UDP of a typical Golgi GT).

- The explanation of why the GT47 domain is inactive lacks biochemical validation. The authors should demonstrate that this domain cannot bind donor substrate in contrast to active GT47 domains of other proteins.

We have added additional data to respond to this point. First, we note that the activity of EXTL3 has been biochemically assayed *in vitro* in at least two previous publications, both of which reported a lack of GlcAT-II activity (Kim et al., 2001 doi: 10.1073/pnas.131188498; Busse et al., 2007 doi: 10.1074/jbc.M703560200; GlcAT-II activity being the only known activity of exostosin GT47 domains). By contrast, the GT47 domain of EXT1 has been shown to exhibit GlcAT-II activity (reviewed by Busse et al., 2014 doi: 10.1016/J.MATBIO.2013.10.001), and the biochemical activity of plant GT47 enzymes

IRX10 and IRX10L have been demonstrated *in vitro* (Urbanowicz et al. 2014 doi: 10.1111/tpj.12643; Wang et al. 2022 doi: 10.1038/s41477-022-01113-1). Our MS aimed to explain these previous biochemical results. We believe that our structure in the presence of 10 mM UDP, in which no UDP density is seen in EXTL3's GT47 domain, already provides significant evidence to support our conclusion.

Furthermore, having further considered this suggestion, we believe that lack of binding would be very difficult to demonstrate by other methods. For example, a lack of any temperature change upon incubation with substrate, as could be measured by techniques such as ITC, could also indicate an isothermal binding reaction, or inappropriate experimental conditions.

Instead, using our highly sensitive *in vitro* assay (as presented in the manuscript), we have tried to demonstrate more clearly that the EXTL3 GT47 domain lacks GlcAT-II activity. We have managed to show that, although present, the GlcAT-II activity in our preparations is much lower than the GT64 domain's GlcNAcT-II activity, which is itself thought to be a minor activity compared with the GT64 domain's main GlcNAcT-I activity (Figs. 1c, Fig. 2, and Supplementary Fig. 1; lines 113–163). Furthermore, by targeting EXT1 in the expression host by CRISPR-Cas9, we have now demonstrated that a major part of the detected GlcAT-II activity is most likely background activity from other minor contaminating exostosins (Fig. 2 and Supplementary Figs 2–3; lines 137–177).

We would like to mention that we have also invested significant efforts to express and purify wild-type and mutant versions of EXT1 in order to clarify the active site of GT47 domain. For these experiments we created 4 different EXT1 expression constructs by two different approaches (His-tag and FLAG-tag vectors) where we introduced EXTL3-like features into the active site of EXT1's GT47 domain. Unfortunately, we were unable to isolate EXT1 protein to our experiments.

- The discussion section is mostly a repetition of the results. It lacks a discussion of what their results offer in the context of EXTL3 being a membrane protein. How would the GT domains be oriented relative to the membrane?

We thank the referee for this suggestion. The discussion has been modified to reduce repetitive elements. We have also expanded the discussion of the role of the GT47 and coiled coil domains in the context of EXTL3 being a membrane protein (lines 386–396). The catalytic part of EXTL3 is connected to the membrane by a long and partly flexible stem domain; hence, the GT domains are not thought to be closely associated with the membrane. Nevertheless, lines 389–391 now clearly indicate the position of the GT domains with respect to the membrane.

Minor points:

- Mainly being a structural work, the authors should provide the pipeline of the cryo-EM data processing.

Pipelines have now been added (Supplementary Figs. 4 and 11).

- The authors should state that there are five different exostosin genes annotated in mammals and described why they focused on EXTL3 so that it is clear for a broader readership.

This is now explained in the introduction (lines 41–44, 48–49, 58–64, and 72–76).

- Panel b in Fig2 is repeated in Fig.3 panel b. This view of the enzyme is constantly repeated in several figures.

We regret the previous lack of clarity in these figures. Though similar, these figures are not repetitions: Panel 3b (previously 2b) shows EXTL3 by itself, whereas panel Fig. 4b (previously 3b) is an alignment of EXTL3 and EXTL2. Nevertheless, the side view has been removed, as the top view should suffice. The colour schemes have been modified to make the contrast between EXTL3 and EXTL2 clearer.

- Fig 2: N and C-termini of the two chains in the dimer should be labeled for better understanding. Gray coloring of the interdomain region cannot be easily visualized, and it should be changed to improve the contrast.

N and C termini are now labelled in 3b. The colour scheme has been altered throughout all figures.

- Fig 4: The amino acids in the protein sequence alignment are not numbered.

The relevant panel has now been removed.

- There is a typo in the figure legend of Fig 6. DDD needs to be changed to DxD

Thank you for noticing this; we have now fixed the error.

- The formatting of tables should be consistent in the supplementary file.

The tables are now more uniform in appearance.

- Figure legends need to be self-explanatory. Many figure legends are only titles that do not supply enough information.

Figure legends have been expanded.

Reviewer #2 (Remarks to the Author):

Heparan sulfate (HS) is a glycosaminoglycan (GAG) polysaccharide of alternating α -1,4-linked N-acetylglucosamine (α -GlcNAc) and β 1,4-linked glucuronic acid (β -GlcA). The HS backbone synthesis is catalyzed by enzymes belonging to the exostosin (EXT) family. These enzymes generally have two glycosyltransferase (GT) domains. The N-terminal domain (classified as CaZy family GT47) is thought to catalyze the β -GlcA transfer, whereas the C-terminal domain (classified as GT64) is involved in addition of α -GlcNAc residue. However, detailed knowledge of the dual functional abilities of EXT family members are limited, mainly due to lack of their structural evidence.

In this manuscript, Wilson et al. reported high-resolution cyto-EM structures of soluble form of an EXTL3 homodimer in the apo-form and bound with UDP and Mn²⁺. Based on the obtained EXTL3 structures, the authors provided structural characteristics of the respective GT domains and their functional importance in EXT-mediated HS backbone synthesis. The authors also proposed that EXTs achieve ordered monosaccharide addition by a simple dissociative mechanism.

Experiments are carefully performed and all data presented in this manuscript is convincing. Therefore, this reviewer feels that the manuscript is acceptable in principle, but still requires some revisions.

We thank the reviewer for these supportive comments.

Major points #1

The authors described that several unique features observed in EXTL3 GT47 domain could be the leading causes for its loss of enzymatic activity to transfer GlcA units. Additional experimental data to support this notion should be required. For example, does genetic engineering of EXT1 to have substitution/insertion of characteristic amino acid residues observed in EXTL3 affect enzymatic activity of the recombinant EXT1 protein? Comparison of the modeled structure of EXT1 with that of EXTL3 would also help understand the proposed molecular mechanisms of GT47-mediated enzymatic activities of EXT proteins.

We thank the reviewer for suggesting these experiments, which we agree would be valuable. In line with the suggestion, we made several attempts to express and purify four different variants of EXT1 by two different approaches (His-tag and FLAG-tag), which is part of the reason for the long revision time. As suggested, we created expression constructs to introduce EXTL3-like features into the active site of EXT1's GT47 domain. Unfortunately, despite our best efforts, expression of EXT1 appears not to be straightforward and ultimately we were unable to obtain protein for any of these variants. Nevertheless, in line with the reviewer's suggestion, we were able to compare the EXTL3 structure with the recently released high-accuracy models of EXT1 and EXT2 by AlphaFold, which strengthen our argument by predicting that EXT1 and EXT2 differ from EXTL3 in the aspect that they exhibit a much more canonical nucleotide sugar binding pocket (Supplementary Fig. 9; please see also manuscript text lines 249–262).

Moreover, we also conducted a series of experiments aiming to confirm the absence of any potential GlcAT-II activity in EXTL3, which involved CRISPR/Cas9 targeting of EXT1 in the expression host (Fig. 1c, Fig. 2, Supplementary Figs. 1–3, and Supplementary Table 3; please see also manuscript text lines 113–177). To our surprise, we detected some GlcAT-II activity in our EXTL3 preparations; however, our data indicate that this activity is a) minor in comparison to the GlcNAcT-II activity and b) at least partly dependent on the presence of trace amounts of EXT1 contamination. Combined with the previously published data on EXTL3 activity, these results help to support our conclusion that EXTL3 has lost GlcAT-II activity in comparison to EXT1.

Major points #2

The authors estimated the distance between the GT47 and GT64 active sites in the EXTL3 structure. Based on the results shown in Fig. 6, they speculated the modes of action of EXT family enzymes. Additional figures showing the domain organization of proposed (or simulated) EXT1 structure would strengthen this interpretation.

We appreciate this comment. We have added an AlphaFold-Multimer model of the EXT1/2 heterodimer to support our predictions (Supplementary Fig. 16; lines 295–303).

Minor points #1

Page 5, line 20, To clarify the importance of a pair of intermolecular disulfide bridges between Cys793 and Cys951 in the EXTL3 homodimer formation, this disulfide structure should be shown in the figure.

A small panel illustrating the disulfide has now been added to Fig. 3 (previously Fig. 2).

Minor points #2

Page 8, lines 9 and 16, citations of Fig. 5d and Fig. 5e in the text should be replaced with Fig. 4d and Fig. 4e, respectively.

Thank you for spotting this; the figures have now been renumbered throughout.

Minor points #3

Page 8, line 17, the authors reported N-glycosylated sites of EXTL3 in detail. Their functional relevance in EXTs should also be discussed.

The section on N-glycosylation has been significantly reduced and moved in part to the supplementary information as requested by reviewer 1 (please see lines 184–185 and Supplementary Fig. 7). The N-glycan on Asn790 is now commented on in the discussion (lines 404–406).

Hiroshi Kitagawa

Reviewers' Comments:

Reviewer #1:

Remarks to the Author:

The article "The structure of EXTL3 helps to explain the different roles of bi-domain exostosins in heparan sulfate synthesis" by Wilson, et al has considerably improved since its initial version. The additional data showing that EXTL3 does not display GlcAT-II activity but it came from traces of EXT1 is convincing, and the discussion on the structural basis for GT47 being non-functional is now clearer. Furthermore, the readability of the article is better than before.

There are a few minor points that still need to be addressed:

Figure legend for Supplementary Figures 6, 7, 13, and 13: Please indicated the contour level of the map

Supplementary Figures 6 and 13: Please label the domains in the structure.

Reviewer #2:

Remarks to the Author:

The authors responded adequately to my requests and comments. Therefore, in my opinion the paper is now acceptable for publication.

Hiroshi Kitagawa

Response to reviewers

Reviewer #1 (Remarks to the Author):

The article “The structure of EXTL3 helps to explain the different roles of bi-domain exostosins in heparan sulfate synthesis” by Wilson, et al has considerably improved since its initial version. The additional data showing that EXTL3 does not display GlcAT-II activity but it came from traces of EXT1 is convincing, and the discussion on the structural basis for GT47 being non-functional is now clearer. Furthermore, the readability of the article is better than before.

We thank the reviewer for helping us to improve the manuscript.

There are a few minor points that still need to be addressed:

Figure legend for Supplementary Figures 6, 7, 13, and 13: Please indicated the contour level of the map

Thank you for pointing this out; contour levels have now been added to these figures.

Supplementary Figures 6 and 13: Please label the domains in the structure.

The domains have now been labelled.

Reviewer #2 (Remarks to the Author):

The authors responded adequately to my requests and comments. Therefore, in my opinion the paper is now acceptable for publication.

We thank the reviewer again for their suggestions and are pleased to see our changes are satisfactory.

Hiroshi Kitagawa